# The two-component system ChvGI maintains cell envelope homeostasis in *Caulobacter crescentus*

Alex Quintero-Yanes[1], Aurélie Mayard[1], Régis Hallez[1,2]*

1 Bacterial Cell cycle & Development (BCcD), Biology of Microorganisms Research Unit (URBM), Namur Research Institute for Life Science (NARILIS), University of Namur, Namur, Belgium, 2 WELBIO, University of Namur, Namur, Belgium

* regis.hallez@unamur.be

**Data Availability Statement:** ChIP-Seq and RNA-Seq data have been deposited to the Gene

## Abstract

Two-component systems (TCS) are often used by bacteria to rapidly assess and respond to environmental changes. The ChvG/ChvI (ChvGI) TCS conserved in α-proteobacteria is known for regulating expression of genes related to exopolysaccharide production, virulence and growth. The sensor kinase ChvG autophosphorylates upon yet unknown signals and phosphorylates the response regulator ChvI to regulate transcription. Recent studies in *Caulobacter crescentus* showed that *chv* mutants are sensitive to vancomycin treatment and fail to grow in synthetic minimal media. In this work, we identified the osmotic imbalance as the main cause of growth impairment in synthetic minimal media. We also determined the ChvI regulon and found that ChvI regulates cell envelope architecture by controlling outer membrane, peptidoglycan assembly/recycling and inner membrane proteins. In addition, we found that ChvI phosphorylation is also activated upon antibiotic treatment with vancomycin. We also challenged *chv* mutants with other cell envelope related stress and found that treatment with antibiotics targeting transpeptidation of peptidoglycan during cell elongation impairs growth of the mutant. Finally, we observed that the sensor kinase ChvG relocates from a patchy-spotty distribution to distinctive foci after transition from complex to synthetic minimal media. Interestingly, this pattern of (re)location has been described for proteins involved in cell growth control and peptidoglycan synthesis upon osmotic shock. Overall, our data support that the ChvGI TCS is mainly used to monitor and respond to osmotic imbalances and damages in the peptidoglycan layer to maintain cell envelope homeostasis.

## Author summary

The cell envelope is the first barrier protecting cells from harsh environmental conditions, such as temperature, pH, oxidative and osmotic imbalances. It is also an obstacle for the intake of antibiotics targeting essential cellular processes. Therefore, molecular components and systems responding to cell envelope stress and maintaining cell envelope homeostasis are important targets for drug therapy. Here we show that the two-

Expression Omnibus (GEO) repository with the accession number GSE200466.

**Funding:** This work was supported by the Fonds de la Recherche Scientifique – FNRS (F.R.S. – FNRS) with a Welbio Starting Grant (WELBIO-CR-2019S-05) to RH. A.Q-Y. was supported by a postdoctoral fellowship from the University of Namur (UNamur). R.H. is a Research Associate of F.R.S. – FNRS. https://www.frs-fnrs.be/en/ The funders had no role in study design, data collection and analysis, decision to publish, or preparation of the manuscript.

**Competing interests:** The authors have declared that no competing interests exist.

component system ChvGI, highly conserved in free-living and pathogenic α-proteobacteria, is activated upon osmotic upshift and treatment with antibiotics targeting peptidoglycan synthesis to enhance the transcription of multiple genes involved in cell envelope homeostasis in *Caulobacter crescentus*. We also show that the kinase sensor ChvG displays a dynamic localisation pattern that changes depending on osmotic imbalance. To our knowledge this is the first two-component system reported to change its cellular localisation upon environmental stress.

## Introduction

Two-component systems (TCS) equip cells with a rapid sensing and response mechanism to optimise survival in changing and stressful environments. The first component of canonical TCS, a sensor histidine kinase (HK), autophosphorylates upon input signal detection on a conserved histidine residue using the gamma-phosphoryl group of ATP. Thereafter, the phosphoryl group is transferred from the HK on a conserved aspartate residue of the second TCS component, a cognate response regulator (RR). In most cases, the RR harbours an output domain that binds to DNA upon phosphorylation to either activate or repress transcription of target genes [1–3]. On the other hand, HKs can also exert specific phosphatase activities on their RRs to turn off the response in absence of the input signal [4].

The Chv (**ch**romosomal **v**irulence factor) TCS conserved in α-proteobacteria and composed of the HK ChvG and the RR ChvI, was first reported as a pathogenicity regulator in *Agrobacterium tumefaciens* responding to acid stress [5,6]. A study in *Sinorhizobium meliloti* showed that the ChvG homologue ExoS is negatively regulated by the periplasmic protein ExoR [7]. Further research in both *S. meliloti* and *A. tumefaciens* showed that low pH triggers ExoR proteolysis to derepress ChvGI activity [8,9]. While unlike other α-proteobacteria, ExoR ortholog is absent in the aquatic free-living bacterium *Caulobacter crescentus*, the ChvGI-dependent response to acidic conditions seems nonetheless conserved in *C. crescentus* [10]. Similarly, the ChvI homologue BvrR of the intracellular pathogen *Brucella abortus* is phosphorylated in combination of low pH and nutrient depletion conditions, which mimics host intracellular conditions [11].

In *C. crescentus*, ChvGI activates the expression of the small non-coding RNA (sRNA) ChvR when exposed to acidic stress or DNA damage with mitomycin C, or when cultured in synthetic minimal media [12]. Once produced, ChvR subsequently inhibits translation of the TonB-dependent receptor (TBDR) ChvT [12]. Interestingly, inactivation of *chvIG* sensitises *C. crescentus* cells to vancomycin treatment whereas a *chvT* mutant becomes resistant, which suggests that vancomycin passes through the outer membrane via the TBDR ChvT to reach the periplasm [13]. In addition, a *chvIG* knock-out mutant (Δ*chvIG*) could not propagate in synthetic minimal media when cells were inoculated at low density, while inactivating *chvT* in a Δ*chvIG* background partially restored growth [10]. Although *chvIG* mutants are sensitive to acid stress when grown in minimal media, the primary cause for the growth defect has not been determined.

ChvGI is also known in *C. crescentus* to coordinate its regulation with another TCS, NtrYX, which is under control of the periplasmic protein NtrZ [10]. NtrZ has been only described in *C. crescentus*, while the HK NtrY and RR NtrX are known for regulating multiple processes in α-proteobacteria such as nitrogen metabolism, motility, virulence and cell envelope integrity [14–17]. Stein *et al.* (2021) [10] showed that both ChvI and NtrX networks significantly overlap and these RRs have opposite regulatory functions in minimal media. For instance, while

ChvI acts as a positive regulator of growth, NtrX inhibits growth. Interestingly, repression of growth in synthetic media is caused by unphosphorylated NtrX, so that inactivating *ntrX* in a Δ*chvI* background also partially restored growth. In fact, NtrY acts as a phosphatase over the phosphorylated NtrX (NtrX~P), while NtrZ inhibits NtrY to presumably maintain high NtrX~P levels.

Here we show that *chvIG* mutants are greatly impacted due to osmotic imbalances in minimal media, but also in complex media supplemented with osmolytes.

Although the general stress response (GSR) sigma factor SigT was shown to be activated upon osmotic imbalance [18], we showed that *chvIG* mutants are much more sensitive to lower hypertonic conditions than a *sigT* mutant. In agreement with previous data [10], we also found that the deletion of either *chvT* or *ntrX* partially restored growth in hypertonic conditions. The ChvI regulon was determined using ChIP-seq with polyclonal anti-ChvI antibodies and RNA-seq, which unveiled new targets related to peptidoglycan (PG) synthesis and recycling. We also showed that ChvG-dependent phosphorylation of ChvI and expression of the *chvIG-hprK* operon is induced upon osmotic shock or treatment with antibiotics targeting the PG. Finally, we observed that upon these stressful conditions, the HK ChvG relocates from a patchy-spotty pattern typically displayed by PG-related proteins to midcell. Overall, our results support a critical role for the ChvGI TCS in maintaining cell morphology and cell envelope homeostasis upon stress.

## Results

### High osmolyte concentration in M2G impairs growth of *chvIG* mutants

It was shown previously that *C. crescentus chvIG* mutants failed to grow in minimal media with xylose as sole carbon source (M2X), while these mutants grew similarly to WT cells in complex media (PYE) [10]. Likewise, we observed that mutants inoculated in M2G (with glucose as sole carbon source) in both solid and liquid media were impaired for growth (**Fig 1A and 1B**). To date, the cause for growth impairment in *chvIG* mutants in minimal media has not been determined. We considered other stressful conditions that are present in synthetic minimal media (M2G or M2X) but missing in PYE as causative agents. Indeed, the synthetic minimal media M2G contains a higher concentration of osmolytes than the complex media PYE [19]. Therefore, we assessed the viability of *chvIG* mutants on M2G plates lacking $Na^+$ and $K^+$ salts (**Fig 1A**, M2G 0% $Na^+$, $K^+$). In these conditions the *chvI* mutant grew similarly to WT cells. We also reduced these osmolytes in liquid cultures, and observed that growth was restored in the *chvI* mutant when half (and below) of the regular concentration found in M2G was used (**Fig 1B**, M2G 50% $Na^+$, $K^+$). Furthermore, we observed that adding osmolytes (NaCl, sucrose and those in M2 salts) to complex media impaired growth and decreased viability of a *chvI* mutant (**Fig 1A**, PYEX + 40 mM NaCl, PYE + 6% sucrose; **Fig 1C**, PYE + M2 salts). Additionally, ectopic expression of *chvI* from the xylose-inducible promoter ($P_{xylX}$::*chvI*) in a *chvI* mutant restored growth in PYEX plates supplemented with NaCl whereas expressing *chvI in trans* in a *chvIG* mutant did not (**Fig 1A**). Finally, in agreement with a previous study [10], we found that a phospho-mimetic mutant of *chvI* (*chvI$_{D52E}$*) grew similarly to WT while a phospho-ablative mutant of *chvI* (*chvI$_{D52A}$*) failed to propagate in M2G (**Fig 1D**). Together, our data show that a fully functional ChvGI TCS is required to survive and grow in hyperosmotic environments.

### Inactivating *chvT* or *ntrX* improves fitness of a *chvI* mutant under osmotic stress

Considering that mutations in *chvT* and *ntrX* partially alleviated the viability impairment of *chvI* mutants in minimal media [10], we tested whether *chvT* or *ntrX* inactivation could also

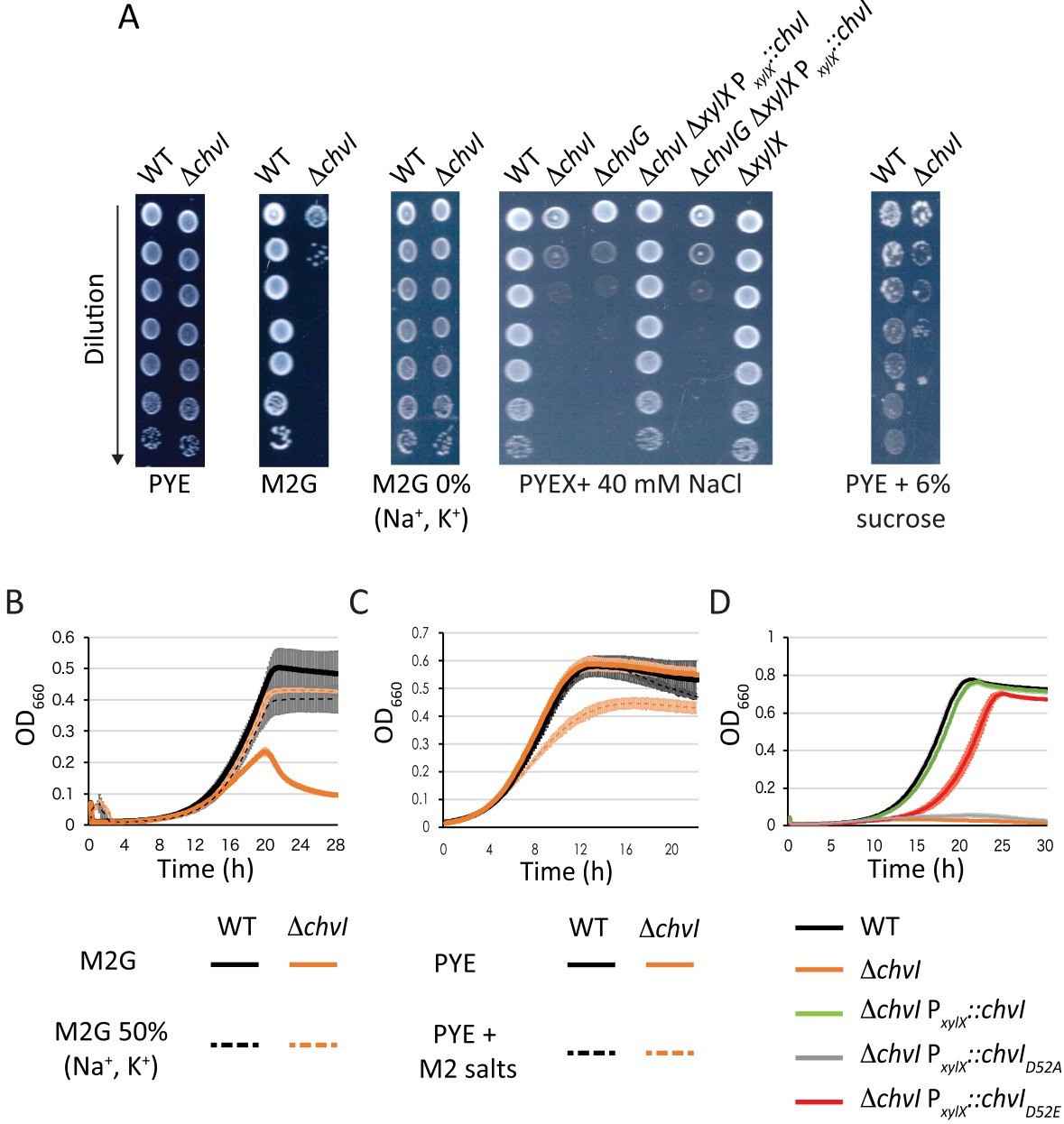

**Fig 1. Viability and growth of *chvGI* mutants in hyperosmotic conditions.** (A) Viability of WT and Δ*chvI* cells in complex (PYE) and synthetic minimal media (M2G) plates with varying osmolytes concentrations. For complementation assays, PYE was supplemented with 0.1% xylose (PYEX). (B) Growth in minimal media with 100% (solid lines) or 50% (dashed lines) Na$_2$HPO$_4$ and KH$_2$PO$_4$ concentrations referred to M2G. (C) Growth in PYE supplemented with 1X M2 salts. (D) Growth in M2G of mutants complemented with WT, phospho-ablative (*chvI$_{D52A}$*) or phospho-mimetic (*chvI$_{D52E}$*) mutants of *chvI* expressed from the xylose-inducible promoter P$_{xylX}$. Data from (B) to (D) represent the average value of biological replicates (n = 3, error bars show standard deviation). WT and Δ*chvI* mutants are represented in (B) to (D) in black and orange, respectively.

partially protect *chvI* mutants from hypertonic conditions. We first confirmed that a Δ*chvI* mutant was impaired for growth under osmotic stress by using an excess of KCl in PYE (**Fig 2**, PYE + 50 mM KCl). In contrast, neither *ntrX* nor *chvT* inactivation interfered with viability in PYE + KCl conditions compared to PYE without KCl (**Fig 2**). The *ntrX* gene (previously annotated as CC1743) was suspected to be essential for growth in complex PYE medium but

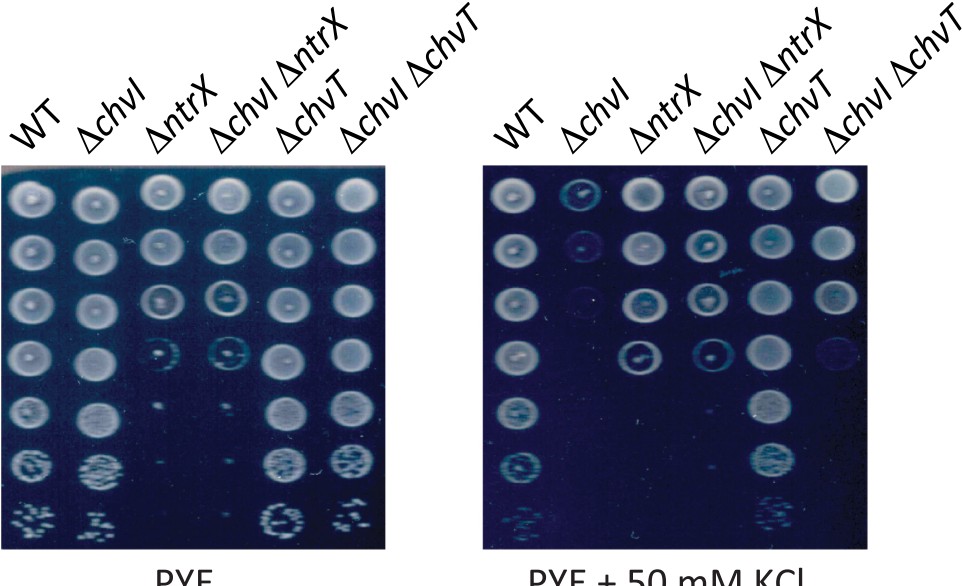

**Fig 2. Viability of *chvI*, *chvT* and *ntrX* mutants in hypertonic conditions.** Viability of single (Δ*chvI*, Δ*chvT* and Δ*ntrX*) and double (Δ*chvI* Δ*chvT* and Δ*chvI* Δ*ntrX*) mutants on PYE agar with or without 50 mM KCl. Images represent three biological replicates.

dispensable for growth in synthetic minimal medium [20]. We found that the viability of *ntrX* mutants was reduced in PYE (**Fig 2**). Notwithstanding this fitness cost, Δ*ntrX* Δ*chvI* double mutants could be generated and grown on PYE, as shown independently in this study and by Stein *et al.* (2021) [10]. More importantly, we observed that inactivating *ntrX* or *chvT* in a Δ*chvI* background partially restored growth in hypertonic conditions (**Fig 2**). These data indicate that additional cell envelope defects to those caused by *chvT* and *ntrX* dysregulation in Δ*chvI* mutants are responsible for growth impairment in hypertonic conditions. Altogether, these data suggest that the homeostasis of the cell envelope is compromised in *chvIG* mutants, leading to a hypersensitivity to osmolytes.

### The ChvI regulon reveals osmotic stress and cell envelope-related target genes

Transcriptomic analyses were previously performed on a Δ*chvI* mutant overexpressing the phospho-mimetic mutant $chvI_{D52E}$ in PYE [10]. However, cells expressing $chvI_{D52E}$ as the only copy, either from its own promoter at the endogenous *chvI* locus ($P_{chvI}::chvI_{D52E}$) or ectopically from the xylose-inducible promoter ($P_{xylX}::chvI_{D52E}$) led to a slight growth delay and cell filamentation (**Figs 1D** and S1). This suggests either that $ChvI_{D52E}$ does not perfectly mimic phosphorylated ChvI or that unphosphorylated ChvI is also required for optimal growth in synthetic minimal media. Hence, we determined the ChvI regulon of WT cells grown in M2G by using ChIP-seq with polyclonal antibodies targeting ChvI, which unveiled the DNA regions directly bound by ChvI (**Fig 3A** and **S1 Table**).

From our experiment, 190 peaks (DNA binding sites) were identified for ChvI (**Fig 3A** and **S1 Table**). Interestingly, the top target is the promoter region of *chvT* itself, suggesting that ChvI regulates ChvT not only indirectly at the post-transcriptional level via transcriptional activation of the sRNA *chvR* but also directly at the transcriptional level (**Fig 3A**). Besides *chvT* and *chvR*, we identified potential new targeted genes involved in (i) cell division, morphology

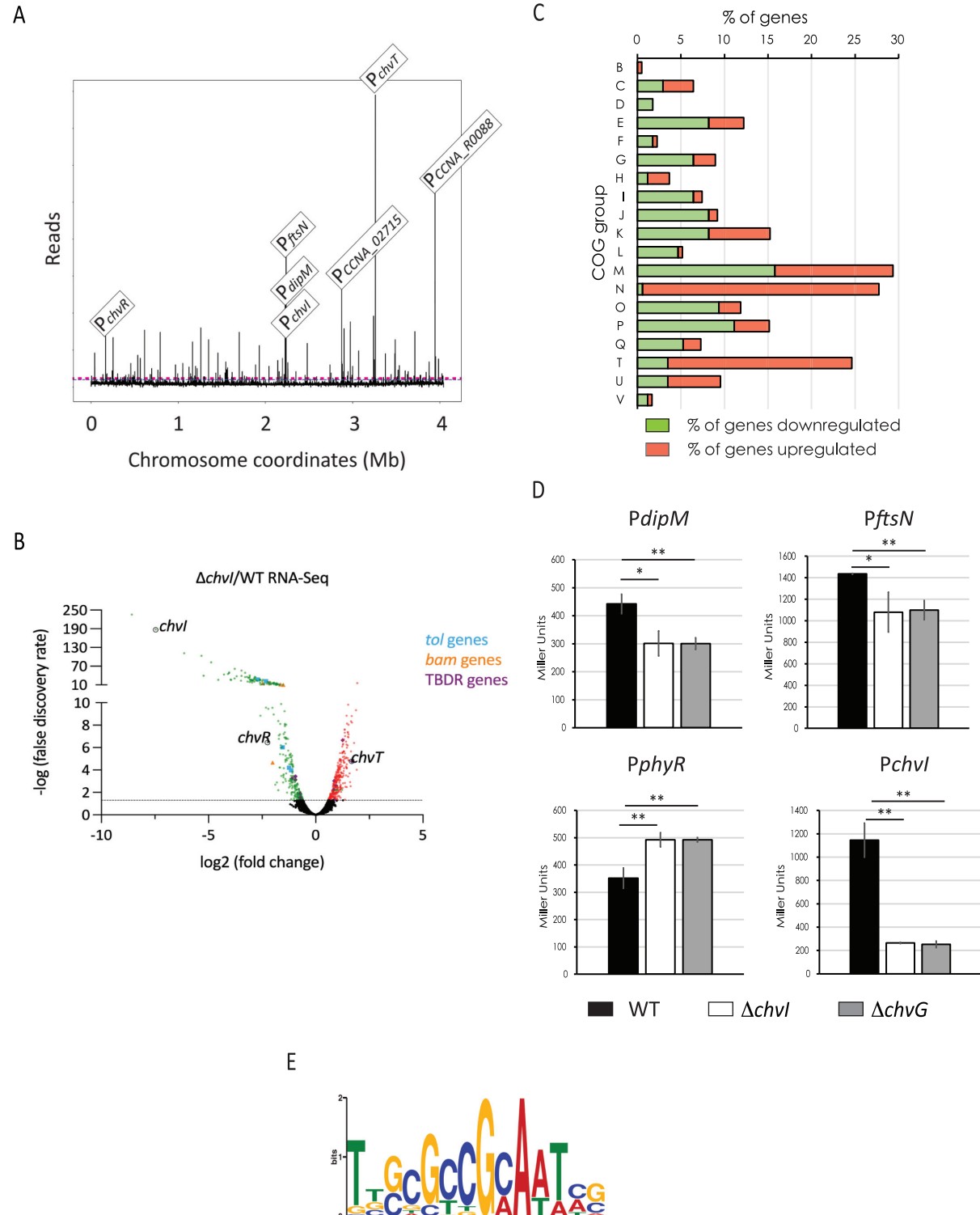

**Fig 3. ChvI target genes determined by ChIP-seq and validated by β-galactosidase assays.** (A) Genome-wide occupancy of ChvI on the chromosome of *C. crescentus* determined by ChIP-seq on WT strain grown in M2G. The x-axis represents the coordinates on the genome (Mb), the y-axis shows the normalized ChIP-Seq read abundance in reads. Some top hits, corresponding to the promoter regions of *chvT*, *CCNA_R0088*, *ftsN*, *CCNA_02715*, *chvR*, *dipM* and *chvI* operon, are highlighted. (B) Volcano plot representing the relation between the fold change and *P* values on gene expression between Δ*chvI* and WT strains exposed to osmotic upshift in PYE 6% sucrose by RNA-seq. Genes

identified are presented as dots. Significant down- and up-regulated genes are presented as green and red dots, respectively, while genes with no significant alterations are presented as black dots. *chvI*, *chvR* and *chvT* genes as well as genes encoding Tol (blue) and Bam (orange) complexes or TonB-dependent receptors (purple) are highlighted. (C) Percentage (%) of genes from the RNA-seq data in the following COG categories: B, Chromatin structure and dynamics; C, Energy production and conversion; D, Cell cycle control, cell division, chromosome partitioning; E, Amino acid transport and metabolism; F, Nucleotide transport and metabolism; G, Carbohydrate transport and metabolism; H, Coenzyme transport and metabolism; I, Lipid transport and metabolism; J, Translation, ribosomal structure and biogenesis; K, Transcription; L, Replication, recombination and repair; M, Cell wall/membrane/envelope biogenesis; N, Cell motility; O, Posttranslational modification, protein turnover, chaperones; P, Inorganic ion transport and metabolism; Q, Secondary metabolites biosynthesis, transport and catabolism; T, Signal transduction mechanisms; U, Intracellular trafficking, secretion and vesicular transport; and V, Defense mechanisms. Genes classified as R, General function prediction only (of them 14% downregulated and 9.5% upregulated); and S, Unknown function (of them 15.8% downregulated and 13.0% upregulated) were not included in the data representation. (D) Activity of the *dipM*, *ftsN*, *phyR* and *chvI* promoters (Miller Units) in WT (black bars), Δ*chvI* (white bars) and Δ*chvG* (gray bars) cells grown overnight in PYE, then washed and exposed for 4h in PYE 6% sucrose. (E) WebLogo of predicted ChvI consensus sequence obtained with MEME. The data in (B) and (C) represent the average values of biological replicates (n = 3, error bars show standard deviation). $^{*} = p < 0.05$, $^{**} = p < 0.01$, single factor ANOVA analysis of β-galactosidase activity.

and peptidoglycan (PG) synthesis, such as *mreB*, *ftsZ*, *ftsN* and *dipM*; and (ii) general stress response (GSR), such as *sigT* (sigma factor T), *nepR* (*sigT* antagonist) and *phyKR* (TCS regulating NepR activity negatively).

Since *chv* mutants cannot grow in synthetic minimal media, we performed RNA-seq on WT and Δ*chvI* cells grown in PYE and exposed to 6% sucrose for 4 hours. We found 301 down-regulated and 293 up-regulated genes with ≥1.5-fold change and a false-discovery rate [FDR] P*adj* ≤ 0.05 (**Fig 3B** and **S2 Table**). Among these genes, 126 (42%) down-regulated and 44 (15%) up-regulated genes were identified in our ChIP-seq experiment done in synthetic minimal media (**S2 Table**).

In agreement with a previous study [10], we confirmed that upon osmotic shock, ChvI regulates the expression of multiple genes involved in cell envelope homeostasis such as for example, *tol* genes, *bam* genes and TBDR-encoding genes (**Fig 3B** and **S2 Table**). Other new targets found in the ChIP-seq experiment were confirmed in the RNA-seq data (**S1** and **S2** **Tables**), such as genes coding for proteins involved in transport, cell-envelope architecture and cell morphology. For instance, among the candidates down-regulated in the *chvI* mutant upon osmotic shock, we found genes encoding the elongasome components MreB, MreC, MreD, Pbp2 and Pbp1a. Also, the FtsZ-binding protein ZapA, the sec-independent protein secretion components TatAB, the type I protein secretion protein RsaD, the moderate affinity potassium transporter Kup, among others. Conversely, among the new candidates up-regulated, we found genes encoding putative TBDRs (CCNA_01034, CCNA_02910, CCNA_03096), among others.

By categorizing the differentially regulated genes found in our RNA-seq data by function with the database of clusters of orthologs groups (COG) (**Fig 3C**), we found that genes classified as "cell wall, membrane, envelope biogenesis" genes (M) were the most represented, with 15,8% and 13,6% of them downregulated or upregulated, respectively. COG analysis also indicates that ChvGI negatively regulates a high number of genes classified in the "motility" (N)– including multiple flagellar and chemotaxis gene components–and "signal transduction" (T) groups (27,1% and 21,1%, respectively).

Then, we performed promoter activity assays for some genes found in RNA-seq and/or in ChIP-seq experiments. First, we could show that ChvGI can act either as a positive regulator (P*dipM*, P*ftsN* and P*chvI*) or a negative regulator (P*phyR*) (**Fig 3D**). These results together with the RNA-seq data further suggest that ChvI, under the control of ChvG, performs direct dual positive and negative transcriptional regulation, but mostly positive. Second, we noticed that the P*dipM*, P*ftsN* and P*phyR* activities in PYE + 6% sucrose (**S2A Fig**) were much lower than those observed in M2G (**Fig 3D**). Therefore, we measured the activity of these promoters in WT

strains upon different osmotic regimes (**S2B–S2D Fig**). Except for $P_{dipM}$ at low osmotic imbalance (PYE + 4% sucrose), we observed that the three promoters significantly increased their activity proportionally to the concentration of osmolytes. Also, we could confirm that exposure to M2G caused the greatest impact on gene expression. However, in addition to the high osmolality of M2G media, it is likely that the transition from PYE (complex media) to M2G (minimal media) increases the stress sensed by the cells.

By using MEME, we searched for a conserved putative binding motif in the promoter regions of genes (i) that were identified in the ChIP-seq and RNA-seq experiments as regulated by ChvI and confirmed in the β-galactosidase assays, and (ii) to which their transcription start sites (TSSs) have been defined experimentally [21,22]. By analyzing a total of 55 sequences (**S3 Table**), we found a GC-rich motif TTGC-$N_3$-GCAA where the 3 central nucleotides mostly being GCC (**Fig 3E**).

## ChvI and SigT respond to different levels of osmotic imbalance

Interestingly, *hprK* is part of the SigT regulon determined upon osmotic shock [23]. Given that *hprK* is part of the same operon as *chvIG*, we tested whether the entire operon *chvIG-hprK* could be under the control of SigT. For that, the activity of the *chvI* promoter ($P_{chvI}$) was measured in cells grown in PYE and exposed for four hours to different osmotic stress. First, we observed that $P_{chvI}$ activity was significantly lower in a Δ*chvI* background in all conditions compared to the WT and the Δ*sigT* mutant (**S3A Fig**). Second, in comparison to the activity measured in unstressed WT and Δ*sigT* cells grown in PYE, $P_{chvI}$ activity strongly increased in both backgrounds when exposed to stressful conditions except for the mild osmotic imbalance PYE + 4% sucrose. Third, $P_{chvI}$ activity was significantly higher in the Δ*sigT* mutant compared to WT when the osmotic imbalance was high, that is in PYE + 6% sucrose, PYE + 85 mM NaCl and M2G (**S3A Fig**). Thus, taking all our observations together suggests that ChvI is subjected to a positive feedback loop while SigT tempers the increase of $P_{chvI}$ activity upon strong osmotic stress.

To better understand the regulatory interplay between ChvI and SigT we tested viability of *chvI* and *sigT* mutants in different osmotic conditions, including those tested in a previous study, such as PYE supplemented with either 85 mM NaCl and 150 mM (5.1%) sucrose [18] (**S3B Fig**). We observed that, contrary to *chvI*, *sigT* is not essential for viability upon low to mild osmotic stress. However, at higher osmotic regimes (PYE + 80 mM NaCl and PYE + 75 mM KCl), the viability of Δ*sigT* mutant was weakly impacted (**S3B Fig**). Moreover, inactivating *sigT* in a Δ*chvI* background did not aggravate the poor viability of Δ*chvI* mutant upon osmotic stress. Together, these data suggest that ChvGI is much more sensitive to osmotic stress than SigT.

## ChvI is phosphorylated under osmotic upshift

Since both single Δ*chvG* and Δ*chvI* mutants are sensitive to osmotic upshift (**Fig 1A**), this suggests that ChvI might be phosphorylated and activated by ChvG in such stressful conditions. We tested this hypothesis by determining the *in vivo* phosphorylated levels of ChvI (ChvI~P) with or without osmotic shock. In order to facilitate uptake of [γ-$^{32}$P] ATP, *Caulobacter* cells were grown in a medium depleted for the phosphate salts $Na_2HPO_4$ and $KH_2PO_4$ (M5GG). Hence, $K^+$ and $Na^+$ concentrations in M5GG are respectively 1.01 mM and 1.02 mM, which is much less than in M2G (7.75 mM of $K^+$ and 12.25 mM of $Na^+$). At such low concentrations of osmolytes, the Δ*chvI* and Δ*chvG* strains grew, as expected, similarly to WT in M5GG (**Fig 4A**). In contrast and in agreement with our previous findings, the growth of the Δ*chvI* mutant was severely impaired in M5GG supplemented with 6% sucrose compared to WT (**Fig 4A**).

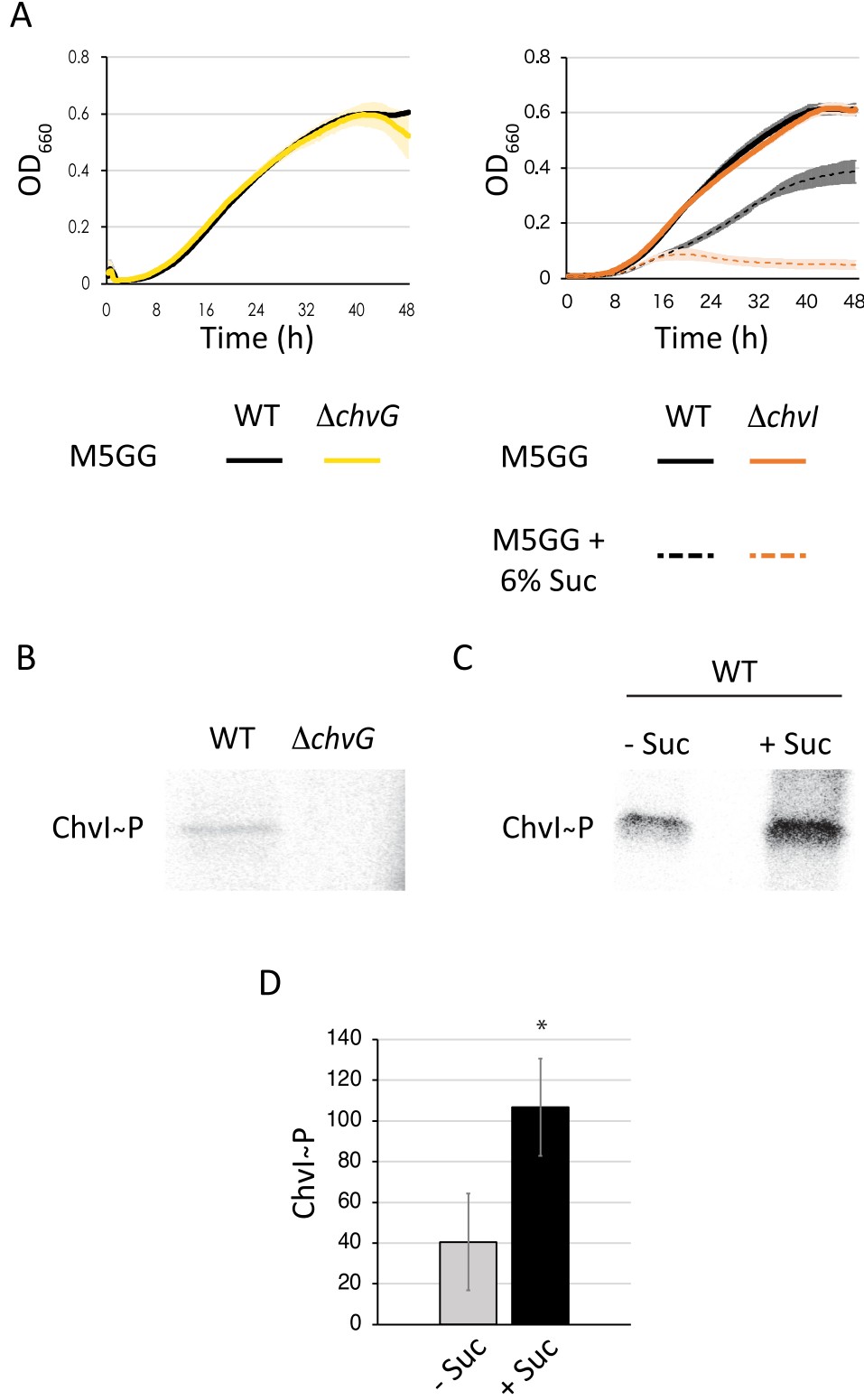

**Fig 4. ChvG-dependent phosphorylation of ChvI is stimulated upon osmotic shock.** (A) Growth of WT (black) and Δ*chvG* (yellow) in M5GG, WT and Δ*chvI* (orange) mutants in M5GG with (dashed lines) or without (solid lines) 6% sucrose (Suc). The data represent the average value of biological replicates (n = 3, error bars show standard deviation). (B) *In vivo* phosphorylation levels of ChvI in WT and Δ*chvG* mutant grown in M5GG. (C) *in vivo* phosphorylation levels of ChvI in WT grown in M5GG exposed (+) or not (-) to sucrose (Suc) for 7 min. (D) Quantified *in vivo*

phosphorylation levels of ChvI in the same growth conditions than in (C). The data represent the average values of biological replicates (n = 3, error bars show standard deviation). * = $p < 0.05$, single factor ANOVA analysis of ChvI phosphorylation.

Consistent with these data, after incubating cells grown in M5GG with [γ-$^{32}$P] ATP for a few minutes and immunoprecipitating ChvI with anti-ChvI antibodies, we found that ChvI was barely phosphorylated in a WT background whereas no ChvI~P was detected in ΔchvG (**Fig 4B**). More importantly, ChvI was hyperphosphorylated when cells grown and incubated in same conditions were exposed a few minutes to osmotic shock with sucrose (**Fig 4C and 4D**). Together these data indicate that the ChvG-dependent phosphorylation of ChvI is indeed enhanced upon osmotic stress.

## ChvGI is sensitive to peptidoglycan synthesis inhibition

Stress response systems surveying cell envelope homeostasis in Gram-negative bacteria are sensitive to different stressful conditions including osmotic shock [24–26]. It has been also shown that deletion of the *chvIG-hprK* operon results in sensitivity to vancomycin [13], which targets the D-Ala-D-Ala moiety of the PG precursors thereby inhibiting PG crosslinking. We confirmed that vancomycin impairs growth and viability of the single ΔchvI mutant (**Fig 5A and 5E**). We also tested growth and viability upon treatment with other cell envelope stressors, such as acidic pH, detergent and antibiotics targeting the outer membrane or the PG synthesis machinery. We did not observe significant differences on fitness between WT and ΔchvI cells under treatment with acidic stress, A22 drug targeting the actin-like protein MreB, SDS and polymyxin B which both target the outer membrane, and the antibiotic cephalexin, which inhibits the PG transpeptidase activity of the Penicillin Binding Protein 3 (Pbp3) associated with the cell division machinery (divisome) (**S4A and S4B** Fig). It is nevertheless noteworthy that the growth curves of WT and ΔchvI cells exposed to polymyxin B treatment (**S4A Fig**) show some differences, but both strains showed also high variability (as seen in error bars) during growth compared to other treatments. In addition, the viability of both strains was similar on plate supplemented with two different concentrations of polymyxin B (**S4A Fig**). In contrast, exposure to other antibiotics targeting PBPs associated to cell elongation machinery (elongasome) showed impaired growth and viability of the ΔchvI mutant compared to WT (**Fig 5A–5E**). For instance, treatment with mecillinam, which inhibits PG crosslinking during elongation by binding to Pbp2 significantly impaired growth and viability of ΔchvI cells (**Fig 5B and 5E**). Also, treatment with antibiotics targeting the bifunctional glycosyltransferase/transpeptidase Pbp1a, such as cefsulodin and moenomycin also diminished the growth and viability of the ΔchvI mutant (**Fig 5C–5E**). Altogether, these results indicated that ChvGI is a TCS responding to osmotic upshift but also to PG crosslinking damage. To confirm this, we assessed the phosphorylation of ChvI following exposure to vancomycin or mecillinam antibiotics (**Fig 5F**). Surprisingly, we observed that ChvI phosphorylation increased after vancomycin but not mecillinam treatment. Since we took a single time point after 7 minutes of antibiotics exposure, we cannot however exclude that these antibiotics activate ChvG phosphorylation in different time windows, depending on their specific mode of action.

Since *chvT* inactivation improved the survival of *chvI* mutants upon osmotic stress (**Fig 2**), we also assessed the viability of the double ΔchvI ΔchvT mutant in plates supplemented with vancomycin, mecillinam, cefsulodin and moenomycin (**S4C Fig**). We found that deletion of *chvT* in a ΔchvI background restored viability to WT levels in plates supplemented with vancomycin. In contrast, in plates containing mecillinam, cefsulodin or moenomycin, viability of the double mutant was partially restored. This partial recovery was similar to that upon

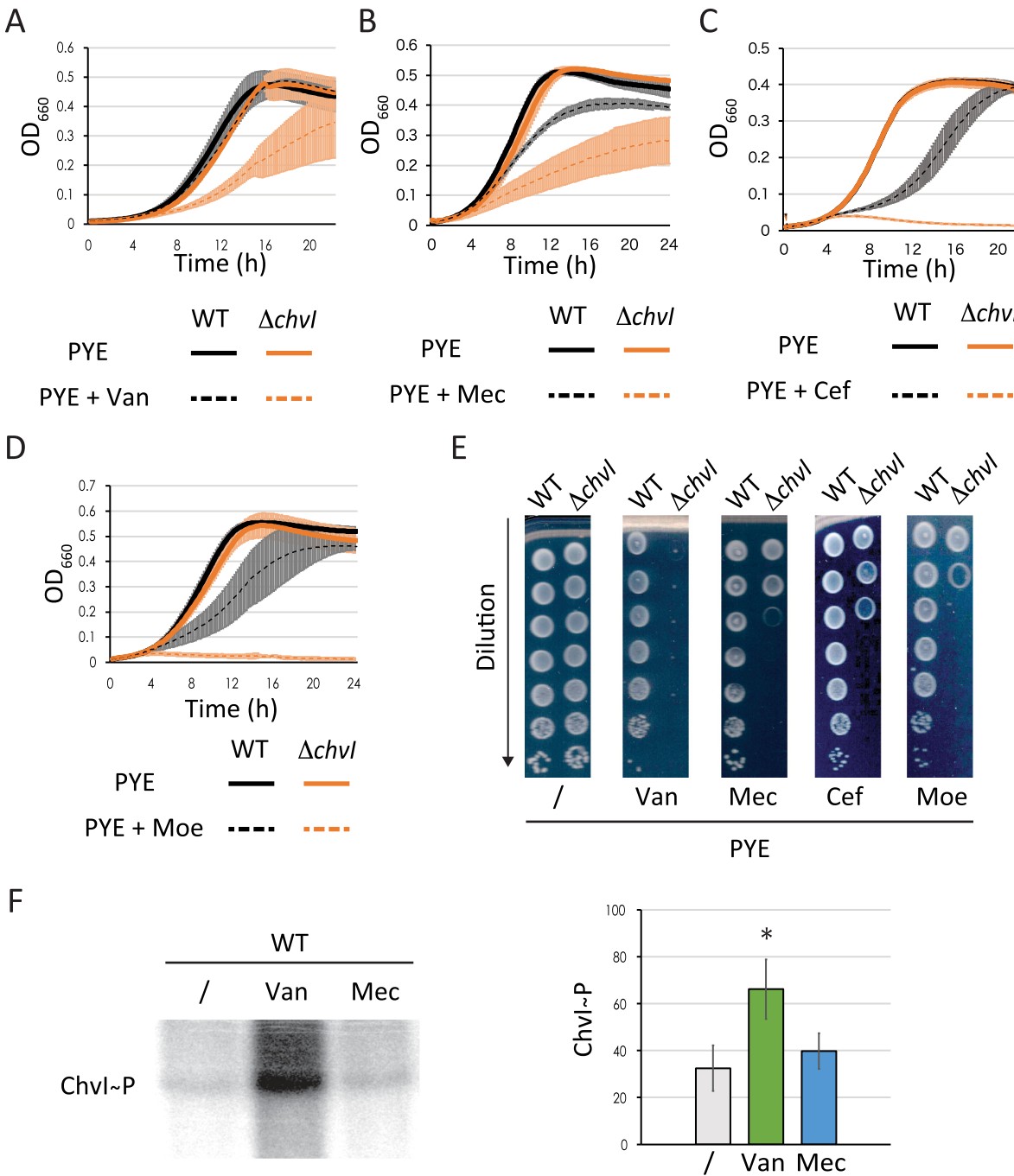

**Fig 5. ChvI responds to treatment with antibiotics targeting transpeptidation of peptidoglycan during elongation** (A-D) Growth of WT and Δ*chvI* mutant cells in PYE supplemented with either vancomycin (Van) (A), mecillinam (Mec) (B), cefsulodin (Cef) (C) or moenomycin (Moe) (D). (E) Viability of WT and Δ*chvI* mutants upon treatment with the antibiotics indicated above. Images are representative of three biological replicates. (F) Quantified *in vivo* phosphorylation levels of ChvI in WT grown in M5GG without antibiotics (/) or with 10 μg/ml vancomycin (Van, green) or 100 μg/ml mecillinam (Mec, blue) for 7 min. The data in figures (A-D) and (F) represent the average values of biological replicates (n = 3, error bars show standard deviation). * = $p < 0.05$, single factor ANOVA analysis of ChvI phosphorylation. WT and Δ*chvI* mutants are represented in (A) to (D) in black and orange, respectively.

treatment with KCl (**Figs 2 and** S4**C**). These data indicate vancomycin is likely imported into the periplasm by ChvT itself while the sensitization of Δ*chvI* cells to osmolytes or other PG-related antibiotics is likely due to the disruption of cell envelope homeostasis following ChvT overproduction in Δ*chvI* cells.

## ChvG relocates from patchy-spotty to midcell upon osmotic shock

Intriguingly, the HK ChvG was previously found in an automated large-scale analysis for protein localisation as displaying a patchy-spotty distribution typically observed with PG-related proteins [27]. Therefore, we constructed fluorescent protein fusions to analyse ChvG and ChvI localisation patterns on PYE and M2G pads. First, we confirmed that eGFP fusions to the N- or C-terminal extremity of ChvG or ChvI were functional and mostly stable, as attested by growth assays and western blot analyses (**S5A and S5B Fig**). Then, fluorescent microscopy images showed that both ChvG N- and C-terminal fusions to eGFP displayed patchy-spotty localisation patterns in cells grown in PYE or M2G (**Figs 6A and S5C**). We also confirmed the patchy-spotty localisation pattern for the ChvG C-terminal fusion to eGFP in high resolution confocal microscopy (**S5D Fig**). Surprisingly, when cells grown in PYE were washed in M2G and imaged on M2G agar pads, we observed that some cells formed foci near mid-cell (**Fig 6A**, **red arrows**). In contrast, cells grown in M2G, washed with PYE and imaged on PYE pad kept their patchy-spotty localisation pattern (**Fig 6A**). This relocation is reminiscent to what was described for the PG-related proteins RodA, Pbp2 and Pbp1a in *C. crescentus*, whose localization changed from a typical patchy-spotty distribution to mid-cell when cells were shifted from PYE to M2G agarose pads [19].

In contrast to ChvG, ChvI fusion to eGFP (ChvI-eGFP) displayed neither patchy-spotty localization nor foci formation after transition from PYE to M2G pads (**Fig 6B**). Instead, the signal was diffused all over the cell body, indicating that the protein remains diffused in the cytoplasm. Then, we wanted to assess if the catalytic activity had any impact on the protein localisation. For that, we fused eGFP to a full-length catalytic mutant of ChvG (ChvG$_{H309N}$), which like Δ*chvG* was unable to grow on M2G (**S5E and S5F Fig**). We observed that, similar to the WT ChvG-eGFP, the ChvG$_{H309N}$-eGFP showed a patchy-spotty pattern when cells were grown in PYE and relocated at mid-cell when shifted from PYE to M2G (**Fig 6C**). Thus, our data suggest that ChvG probably co-localizes with PG synthesis machinery independently of its kinase activity.

## The N-terminal extremity of ChvG determines localisation and relocation

ChvG is a HK anchored in the membrane thanks to 2 predicted transmembrane helices which delimit a periplasmic sensor domain (amino acids 50–221), with a cytoplasmic signal transduction histidine kinase domain (amino acids 242–534) (**Fig 7A**). To assess which of these domains are essential for localisation, we fused mCherry to ChvG versions harbouring either the complete sensor (ChvG$_{1-274}$ and ChvG$_{1-202}$) or the catalytic (ChvG$_{273-534}$) domain. First, we showed that both sensor and catalytic domains are essential for *C. crescentus* to propagate in high osmolytes conditions like in M2G since mutants harbouring in frame deletion of each of these domains (Δ*chvG$_{274-534}$* or Δ*chvG$_{1-274}$*) were unable to survive in M2G (**S5G Fig**). The expression of the full length ChvG fused to mCherry in the Δ*chvG* mutant restored growth in M2G whereas truncated ChvG portions fused to mCherry did not (**S5H Fig**). Then, we observed that full-length ChvG fused to mCherry displayed similar localisation and relocation patterns (**Fig 7B and 7C**, **green arrows**) than the ones described for eGFP fusions (**Fig 6A**). Interestingly, the complete sensor domain alone (ChvG$_{1-202}$, ChvG$_{1-274}$) fused to mCherry were both localized as patchy-spotty in PYE (**Fig 7B**). However, only the full-length ChvG-

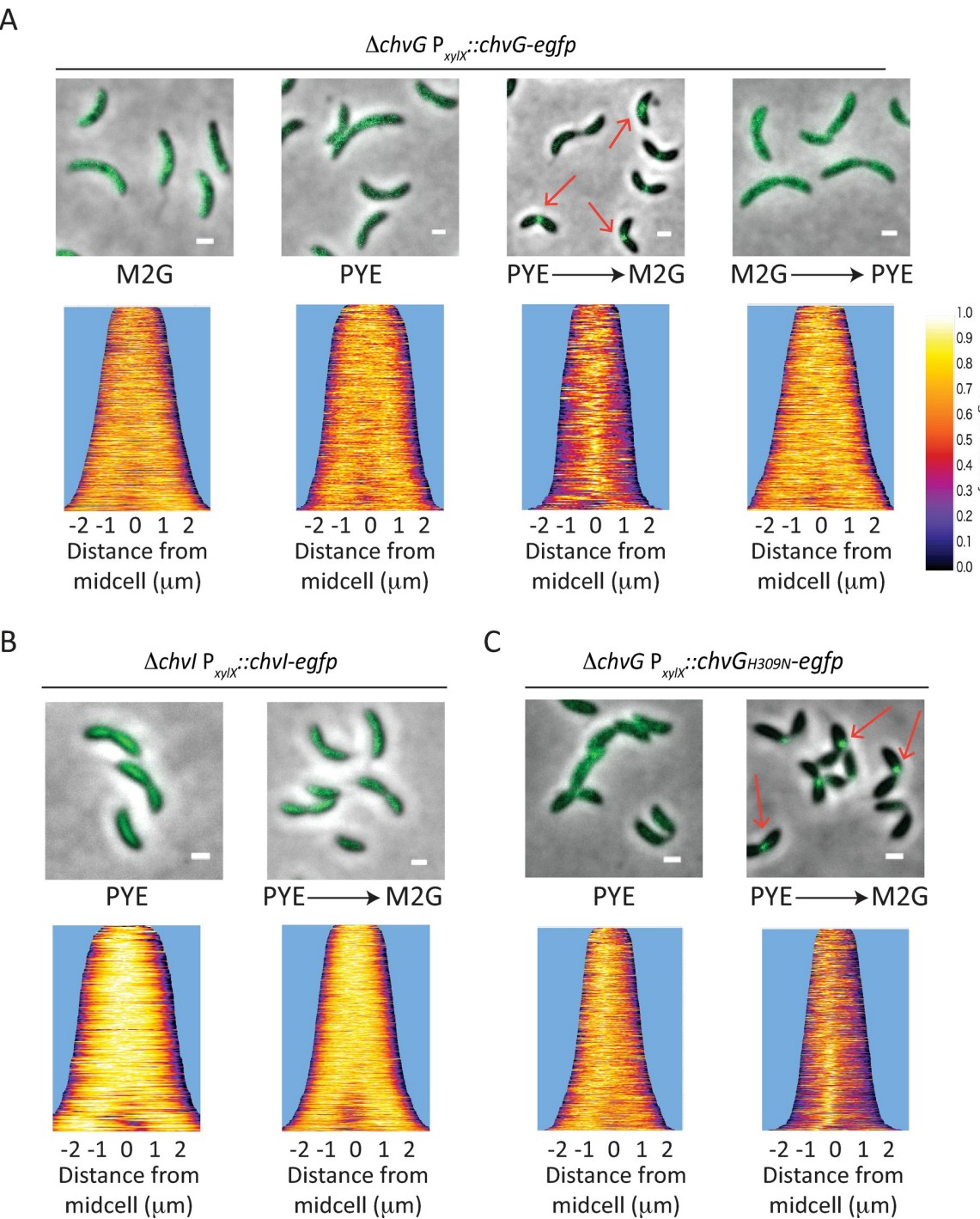

**Fig 6. ChvG relocates from a patchy-spotty pattern to mid-cell upon osmotic upshift.** (A) Localisation of ChvG-eGFP in cells grown in either M2G or PYE and imaged on M2G or PYE agar pads, respectively; or cells grown in PYE, washed in M2G and imaged on M2G agarose pads (PYE ➔ M2G); or cells grown in M2G, washed in PYE and imaged on PYE agarose pads (M2G ➔ PYE). (B) Localisation of ChvI-eGFP in a Δ*chvI* background and (C) ChvG$_{H309N}$-eGFP in a Δ*chvG* background grown overnight in PYE and imaged on PYE agarose pads or grown in PYE, washed in M2G and imaged on M2G agarose pads (PYE ➔ M2G). Demograph data represent cells sorted from short to long length with no less than 200 cells per sample. Liquid cultures and pads were supplemented with 0.1% xylose to allow expression of ChvG, ChvI and ChvG$_{H309N}$ fused to eGFP fusions. Scale bar = 1 μm.

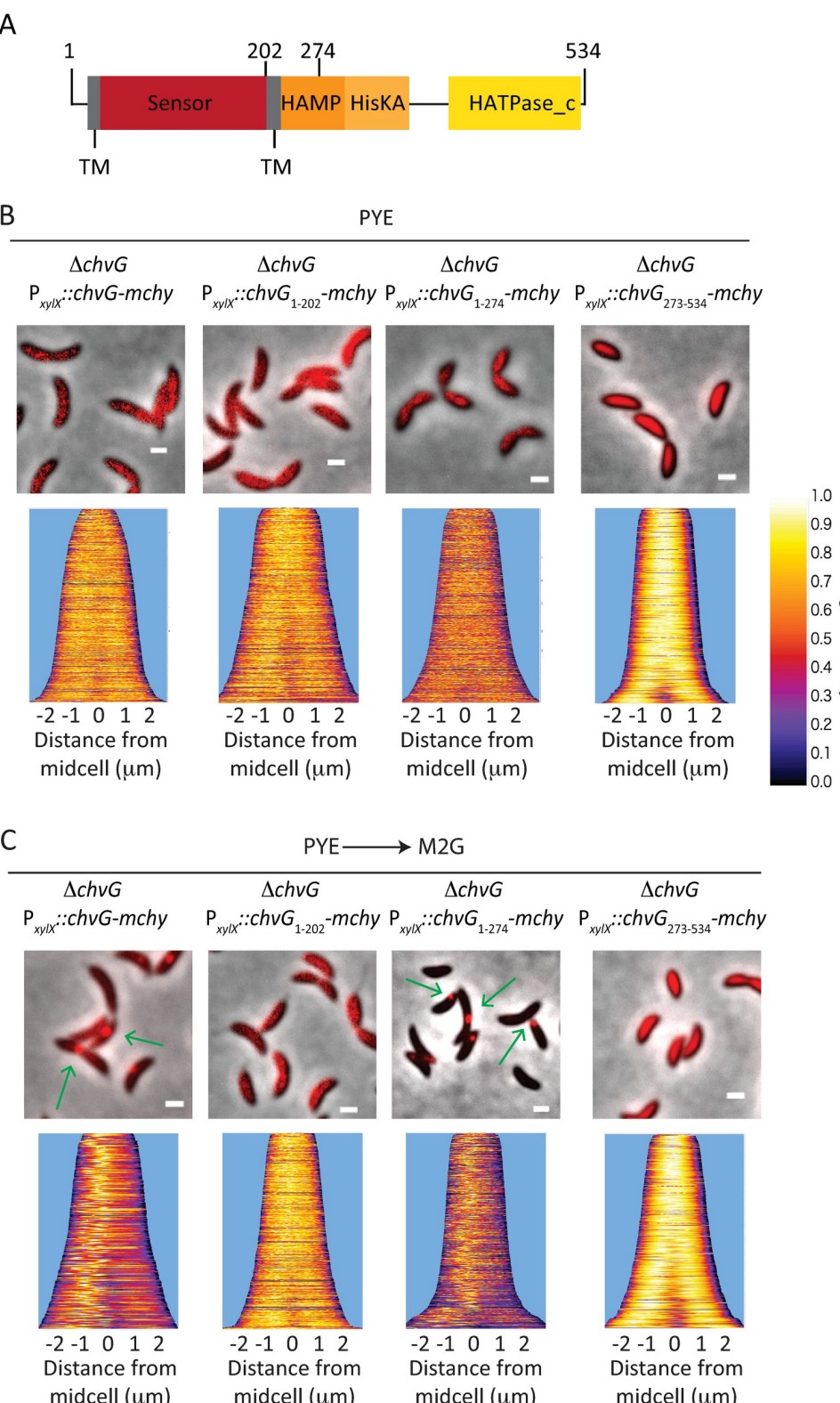

**Fig 7. The N-terminal periplasmic sensor domain of ChvG is required for patchy-spotty localisation pattern and mid-cell relocation.** (A) Conserved domains in the HK ChvG and position of the amino acids defining the truncated proteins. (B-C) Localisation of ChvG WT and truncated versions in cells grown in PYE and imaged on PYE agarose pads (B) or in cells grown in PYE, washed in M2G and imaged on M2G agarose pads (PYE ➔ M2G) (C). Liquid

cultures and pads were supplemented with 0.1% xylose. Demograph data represent cells sorted from short to long length with no less than 300 cells per sample. Scale bars in microscopy images correspond to 1 μm.

mCherry and ChvG$_{1-274}$-mCherry fusions relocated as foci upon transition to M2G, even though ChvG$_{1-274}$-mCherry formed foci that were mostly not found at midcell (**Fig 7C, green arrows**). Unlike the sensor domain truncates, the catalytic domain (ChvG$_{273-534}$) fused to mCherry displayed neither the patchy-spotty nor the foci pattern of localization, but was instead diffusely localized in the cytoplasm (**Fig 7B and 7C**). Altogether, our results indicate that the periplasmic sensor domain of ChvG is sufficient to determine the patchy-spotty pattern of localization but not to relocate as foci upon osmotic shock.

## Discussion

The α-proteobacterial TCS ChvGI has conserved cell envelope regulatory functions [10,13,28–30], but the exact perceived signals that trigger the ChvG-dependent phosphorylation of ChvI remained unknown. Our study shows that hyperosmotic shock and antibiotic treatment harming PG crosslinking stimulate ChvI phosphorylation. Also, we showed that other antibiotics targeting PG synthesis impair fitness of the Δ*chvI* mutant. On top of this, we showed that ChvI regulates expression of genes related to PG synthesis, such as *ftsN* and *dipM*, and to osmotic and oxidative stress regulation, such as *nepR*, likely explaining why ChvGI system is required for viability in these stressful conditions. Other targets related to cell envelope architecture, identified by ChIP-seq and RNA-seq experiments, include genes coding for the β-barrel assembly machinery (Bam) complex; components of the Tol system; several outer membrane proteins (OMP) such as RasFa, CCNA_03820 and CCNA_01956, and several TBDR proteins, protein export systems; elongasome and divisome-related proteins such as MreB, MreC, MreD, Pbp1a and the cytoplasmic FtsZ-binding protein ZapA. Altogether, our results support a role for ChvGI in *C. crescentus* as a safeguard of cell envelope homeostasis by sensing cell wall-related damages and regulating the expression of genes related to cell division and envelope architecture.

Categorization of the genes identified in the RNA-seq experiments based on their function using the COG database confirmed that ChvGI regulates both positively a negatively a large number of genes involved in cell envelope biogenesis. Moreover, this analysis indicates that ChvGI is also an important regulator of genes related to motility and signal transduction mechanisms. Although ChvGI homologues were initially studied in the context of acid-dependent induction of virulence [5,9,11], transcriptomic data from other α-proteobacteria also indicate that ChvI is a conserved cell-envelope regulator. For instance, the ExoR-dependent regulon determined in *A. tumefaciens* also revealed that genes related to cell envelope, motility and chemotaxis were among the top ChvI-regulated genes [28]. Other studies on ChvI homologues in *B. abortus* and *S. meliloti* also found several genes related to PG, outer and inner membranes architecture [29,31] as well as to motility, stress response and transport [31,32]. Taken together, this suggests ChvGI evolved as a critical cell-envelope safeguard system in α-proteobacteria.

A putative DNA binding consensus sequence for ChvI (TTGC-N$_3$-GCAA, with N$_3$ being most often GCC) was found thanks to the analysis of the upstream sequences of genes found (i) in the ChIP-seq experiment to be bound by ChvI and (ii) in the RNA-seq and the promoter activity assays to be regulated by ChvI. Interestingly, a consensus ChvI binding sequence was previously reported in *S. meliloti* as a 15bp long sequence harbouring GCC direct trinucleotides repeats (N-GCC-N$_8$-GCC) [30]. However, although our sequence analysis revealed

reasonably conserved DNA sequences that could be recognized by the ChvI protein, experimental validation will be required.

In the absence of a functional ChvGI system, *C. crescentus* cannot propagate in minimal media except if either the TBDR ChvT or the RR NtrX is concomitantly inactivated [10]. ChvGI is known to indirectly down-regulate ChvT by directly activating the expression of the sRNA ChvR [12]. Our ChIP-seq data showed that ChvI also likely regulates *chvT* expression directly by binding to its promoter region (**Fig 8**). This dual–transcriptional and post-transcriptional–control indicates that ChvT is an important player of cell envelope homeostasis in *C. crescentus*. Interestingly, NtrYX has been also described as a cell envelope regulator in α-proteobacteria. NtrYX controls succinoglycan and exopolysaccharide (EPS) production, and salt stress response in *S. meliloti* [17,33]. In *Rhodobacter sphaeroides*, NtrYX confers resistance to membrane disruptive agents and regulates the expression of genes coding for PG and EPS synthesis enzymes, lipoproteins and cell division proteins [15,34]. Thus, confirming that *ntrX* inactivation in a Δ*chvI* background partially restored growth in the presence of high concentration of osmolytes further supports the importance of the NtrZXY system, together with ChvGI, in the cell envelope stress response in α-proteobacteria.

In *C. crescentus*, the general stress response (GSR) sigma factor SigT is also activated upon osmotic imbalance thanks to a complex network comprising the sRNA GsrN, the anti-SigT regulator NepR, the histidine phosphotransferases LovK and PhyK and the RR MrrA, LovR and PhyR [18,23,35–37]. In steady-state conditions, NepR impedes SigT-dependent transcription. Upon either oxidative stress or osmotic imbalance, the histidine phosphotransferase MrrA activates PhyK to further phosphorylate PhyR. Once phosphorylated, PhyR~P interacts with NepR to release SigT from inhibition and allow the expression of the SigT-dependent regulon. Interestingly, we found that ChvGI and GSR are interconnected but this connection is counterintuitive. Indeed, we observed that deletion of *sigT* led to higher *chvI* promoter activity and that ChvI directly represses *phyR* expression, suggesting that ChvI and SigT antagonise each other while being sensitive to hyperosmotic conditions. However, we cannot exclude the possibility that the cell envelope homeostasis is sufficiently disrupted in both single mutants to correspondingly activate the other functional system, *i.e.* ChvGI in Δ*sigT* through P$_{chvI}$ and SigT in Δ*chvI* through P$_{phyR}$. The antagonistic regulation might allow the two systems to respond to different levels of osmotic imbalances. For instance, activation of ChvGI at low hypertonic conditions could down-regulate GSR whereas SigT would be sensitive to higher salt concentrations to which ChvGI is turned down. In agreement with this hypothesis, we found that SigT is required in more severe osmotic scenarios compared to ChvI. Considering that SigT is a sigma factor, it is expected that SigT negatively regulates *chvI* expression indirectly.

In addition, constitutive activation of a cell envelope stress response system can be detrimental. Overexpression of a phosphomimetic *chvI* variant has been shown to complement growth, but also to result in dysfunctional filamentous growth in minimal media [10]. The importance of negative regulation of cell envelope stress response systems has been lucidly demonstrated in *E. coli* with the sigma factor E (σ$^E$), the inner membrane stress regulator Cpx or the regulator of capsule synthesis Rcs [38–41]. Indeed, overactivation of σ$^E$ causes overexpression of sRNAs inhibiting expression of integral OMPs, therefore weakening the outer membrane integrity [42]. Likewise, constitutive Cpx activation disturbs cell division and morphology causing mis-localisation of FtsZ as well as overexpression of L,D-transpeptidase enzyme LdtD [41,43]. Similarly, overactivation of the RR RcsB results in overexpression of the small outer membrane lipoprotein OsmB, which is toxic through yet unknown mechanisms [39]. In any of the previous cases, hyperstimulation can lead to deregulation of cell envelope components.

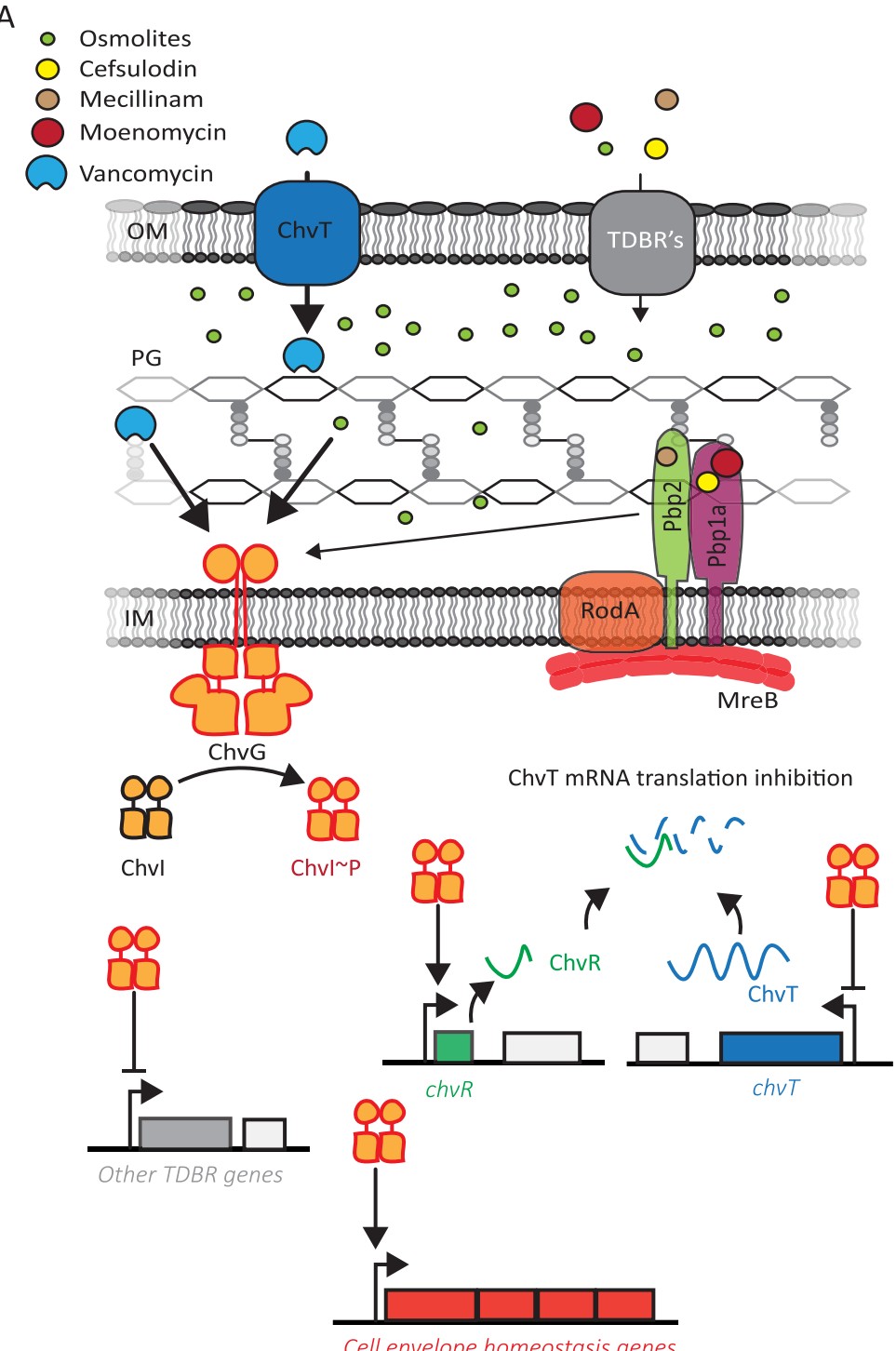

**Fig 8. Model of ChvGI response to cell-envelope stressors.** ChvT and other TDBR's sensitize *C. crescentus* by transporting antibiotics targeting PG transpeptidation and osmolytes into the periplasm. Then, the HK ChvG autophosphorylates and activates the RR ChvI to upregulate the expression of genes related to cell envelope homeostasis and down-regulate outer membrane transport (such as TDBR).

We showed that ChvGI also regulates sensitivity to the antibiotic mecillinam, cefsulodin and moenomycin which all inhibit PG transpeptidation (**Fig 8**). Previously, Vallet *et al.* (2020) [13] showed that a mutant for the *chvIG-hprK* operon failed to grow when exposed to vancomycin, which also impedes PG crosslinking by directly interacting with the D-Ala-D-Ala moiety of the PG precursors (**Fig 8**). Here, we showed that ChvI was phosphorylated when exposed to vancomycin but not to mecillinam. Although both antibiotics (i) inhibit the transpeptidation step of PG synthesis and (ii) dramatically impact the viability of ΔchvI cells, they might induce ChvG phosphorylation in different time windows. This could be explained if these antibiotics cross the outer membrane with different efficiency. Consistent with this idea, our data suggest that vancomycin is efficiently transported by the TBDR ChvT, while other antibiotics (i.e. cefsulodin, mecillinam and moenomycin) are either poorly transported by ChvT or mostly transported by other OMPs (**Fig 8**). It is possible that these antibiotics reach toxic concentrations in the periplasm at different times. It is therefore likely that ChvGI is systematically activated upon inhibition of PG transpeptidation during cell elongation but with different sensitivity depending on the transport rate for each antibiotic. The multiple genes encoding TDBRs identified in the ChIP-seq and RNA-seq experiments support the idea that down-regulation of the outer membrane transport is a key strategy in the ChvGI-dependent cell envelope safeguard (**Fig 8**).

In contrast to the stressors mentioned above, treatment with cefixime, cefotaxime and sodium deoxycholate did not show differences in cell viability between WT and ΔchvIG-hprK cells [13]. Furthermore, the activation of *chvR* expression upon cefotaxime or sodium deoxycholate exposure was shown to be ChvI-independent. In addition, we showed here that ΔchvI cells were not more sensitive than WT cells to treatment with SDS, polymyxin B or the MreB inhibitor A22. Considering that MreB is a key cytoplasmic component of the elongasome complex, of which Pbp2 and other proteins are part, it is therefore likely that ChvGI rather senses specific targets in the periplasm to assess cell envelope stress (**Fig 8**). We propose that ChvG phosphorylation is stimulated by hyperosmotic conditions or antibiotics targeting PG transpeptidation by specifically sensing the osmotic pressure or the integrity of the peptidic crosslinks of PG instead of glycan strands. In such scenario, the periplasmic sensor domain of ChvG could interact with components of the elongasome or bind peptide bridges of the PG. Notwithstanding that, other stress response systems, yet to be discovered or characterised in the context of the aforementioned stressors, might be involved in responding to the damages that do not activate ChvGI.

We observed that ChvG has a patchy-spotty localisation pattern when grown in complex and synthetic minimal media, while it relocates as foci near mid-cell upon transition from complex to hyperosmotic minimal media. The patchy-spotty localisation pattern has been described for multiple PG-related proteins in *C. crescentus*, including MreB, MreC, Pal, Pbp1a, Pbp2, RodA, TolB [19,27,44–46]. Moreover, RodA, Pbp2, Pbp1a, which are proteins involved in PG polymerization and crosslinking, have been reported to relocate at mid-cell upon osmotic upshift in a FtsZ-dependent but MreB-independent manner [19]. It would be interesting to assess whether this ChvG localisation patterns are conserved among α-proteobacteria, in particular in Rhizobiales since they do not encode MreB orthologs [47,48]. In *E. coli*, it has been reported that the Pbp2 periplasmic portion physically interacts with those of Pbp1a and RodA to mediate PG assembly and to ensure proper cell elongation [49,50]. It is yet to be determined whether ChvG interacts with Pbp2, Pbp1a, RodA and/or others with similar localisation patterns. Nonetheless, it is tempting to speculate that ChvG interacts and co-localises with PG-related enzymes as a cell envelope safeguard system in α-proteobacteria. A recent study in the Gram-positive bacterium *Bacillus thuringiensis* showed that upon treatment with the antibiotic cefoxitin, the putative PBP protein PbpP derepresses the extracytoplasmic

function sigma factor P ($\sigma^P$) which increases expression of β-lactamases [51]. Also, it has been suggested that PG-related proteins facilitate *C. crescentus* adaptation to hyper-osmotic conditions by relocating to mid-cell (24). Repositioning the PG-crosslinking proteins at mid-cell might support cell-wall synthesis at a position where other proteins important to maintain the cell-envelope scaffold (i.e Tol/Pal complex proteins) and dictate cell-division (i.e. divisome proteins) are located. Recently, it has been shown that treatment with osmolytes and nanoparticles that create hypertonic conditions near the outer membrane result in aggregation of OMPs, such as OmpA and OmpC, in *E. coli* [52]. Interestingly, this osmotic-dependent aggregation inactivates OmpA and OmpC and stimulates persistence to antibiotics that are transported by these OMPs. Therefore, it is possible that the ChvG relocation depends on the aggregation of OMPs as well as on periplasmic PG-associated proteins to maintain surveillance on cell envelope-related damages in the process of adaptation to hyper osmotic environments. Future experiments will aim to determine the ChvG interactome under conditions which compromise the cell envelope, not only in *C. crescentus* but also in other α-proteobacteria, to dissect activation mechanisms as well as its connection with the general stress response.

## Methods

### Bacterial strains and growth conditions

Strains, plasmids and oligonucleotides used in this study are listed in supplementary tables (**S4**, **S5** and **S6** **Tables**). Plasmid construction details are presented in **S1 Text**. *E. coli* strains were grown aerobically in either LB (broth) sigma or LB + 1.5% agar at 37˚C. All *C. crescentus* strains in this study are derived of the NA1000 wild-type strain, and growth was achieved at 30˚C in aerated conditions using either complex medium Peptone Yeast Extract (PYE) or synthetic media supplemented with glucose (M2G or M5GG) as already described in [53]. M2G and M5GG were prepared using M2 (12.25 mM $Na_2HPO_4$, 7.75 mM $KH_2PO_4$, 9.35 mM $NH_4Cl$) and M5 (10 mM PIPES pH 7.2, 1 mM NaCl, 1 mM KCl, 0.37 mM $Na_2HPO_4$, 0.23 mM $KH_2PO_4$) salts, respectively, and both supplemented with (0.5 mM $MgSO_4$, 0.5 mM $CaCl_2$, 0.01 mM $FeSO_4$, 0.2% glucose). 1 mM glutamate sodium was added to make M5GG. Modified M2G 0% ($Na^+$, $K^+$), M2G 25% ($Na^+$, $K^+$) and M2G 50% ($Na^+$, $K^+$) were prepared without $Na_2HPO_4$ and $KH_2PO_4$ or either 50% or 25% of the $Na_2HPO_4$ and $KH_2PO_4$ concentrations in M2G, respectively. *C. crescentus* was grown on plates using either PYE or M2G supplemented with 1.5% agar at 30˚C. Growth was monitored by measuring absorbance at $OD_{660}$ in liquid cultures using an automated plate reader (Biotek, Epoch 2) with continuous shaking at 30˚C. Gene expression under the control of inducible promoter ($P_{xylX}$) was induced with 0.1% D (+)-xylose (Sigma). Expression of enhanced green fluorescent protein (*egfp*) and monomeric cherry (*mCherry*) protein fusions was induced in fresh exponentially growing cultures ($OD_{660}$ ~ 0.1) for 4 h and overnight, respectively. Generalized transduction was performed with phage Cr30 according to the procedure described in [54]. Antibiotics for *E. coli* were used with the following final concentrations (g ml$^{-1}$, in liquid/ solid medium) ampicillin (50/100), kanamycin (30/50), chloramphenicol (30/20), and oxytetracycline (12.5/12.5) for *C. crescentus* A22 (2.5/5) cefsulodin (80/80), cephalexin (15/15), mecillinam (100/100), moenomycin (0.5/1), kanamycin (5/20), oxytetracycline (1/2.5), polymyxin B (5/12.5), vancomycin (10/10) where appropriate. Plasmid delivery into *C. crescentus* was achieved by either bi- or tri-parental mating using *E. coli* S17-1 and *E. coli* MT607 as helper strains, respectively. In-frame deletions were created by using the pNPTS138-derivative plasmids as follows. Integration of the plasmids in the *C. crescentus* genome after single homologous recombination were selected on PYE plates supplemented with kanamycin. Three independent recombinant clones were inoculated in PYE medium without kanamycin and incubated overnight at 30˚C. Then, dilutions

were spread on PYE plates supplemented with 3% sucrose and incubated at 30˚C. Single colonies were picked and transferred onto PYE plates with and without kanamycin. Finally, to discriminate between mutated and wild-type loci, kanamycin-sensitive clones were tested by PCR on colony using locus-specific oligonucleotides.

## Spotting assays

Ten-fold serial dilutions (in PYE) were prepared in 96-well plates from 5 ml cultures in standard glass tubes grown overnight at 30˚C in the corresponding media. Cells (5 l) were then spotted on plates, incubated at 30˚C for two-to-three days and pictures were taken. Cells in assays including strains with mutations in $\Delta ntrX$ were normalised to $OD_{660}$ 1.0 since these mutations lead to growth defects in PYE.

## β-galactosidase assays

β-galactosidase assays were performed as already described in [55]. Briefly, overnight saturated cultures of *Caulobacter* cells harbouring *lacZ* reporter plasmids were diluted to $OD_{660}$ 0.05 in fresh medium and incubated at 30˚C until $OD_{660}$ of 0.3 to 0.5. 100 μl samples were collected in a 96 well plate and kept at -80˚C until measurement. Then, 50 μl of aliquots of the previously frozen samples were thawed and immediately treated with 50 μl Z buffer (60 mM $Na_2HPO_4$, 40 mM $NaH_2PO_4$, 10 mM KCl, 1 mM $MgSO_4$, pH 7.0) supplemented with 0.1 g polymyxin B and 0,27% (v/v) β-mercaptoethanol for 30 min at 28˚C. To this, 150 μl of Z buffer were added, followed by 50 μl of 4 mg/ml O-nitrophenyl-β-D-galactopyranoside (ONPG). Then, ONPG hydrolysis was measured at 30˚C for 30 min. The activity of the β-galactosidase expressed in miller units (MU) was calculated using the following equation: $MU = (OD_{420} \times 1,000) / [OD_{660} \times t \times v]$ where "t" is the time of the reaction (min), and "v" is the volume of cultures used in the assays (ml). Experimental values were the average of three independent experiments.

## Microscopy

Strains grown in PYE were imaged either in exponential or stationary phase using 1.5% agar pads with the indicated medium. Cells in osmotic shock conditions were pelleted and washed twice with the indicated stress medium and imaged in 1.5% agar pads maintaining the stress condition. Images were obtained using Axio Imager Z1 microscope (Zeiss), Orca-Flash 4.0 camera (Hamamatsu) and Zen 2.3 software (Zeiss). Temperature (30˚C) was maintained stable during microscopy analysis using the Tempcontrol 37-analog 1 channel equipment (Hemo-Genix) coupled to the Axio Imager Z1 microscope. Images were processed with ImageJ. Demographs were obtained with MicrobeJ by segmenting each cell, integrating fluorescence, sorting cells by length and plotting fluorescence intensity and cellular widths to indicate the relative position of the protein fusions, in which 0 represents mid-cell and 2 or -2 the cell poles [56]. Confocal microscopy images were obtained using a LSM900 Ayriscan microscope. Samples were prepared for imaging using cells grown to $OD_{660}$ 0.5. Cells were fixed immediately with a 12.5% paraformaldehyde and 150 mM $NaPO_4$ solution. Cells were mixed gently and incubated for 15 min at room temperature, then at 4˚C for 40 min. Then, cells were washed four times with 1.5X volume of $dH_2O$. Cells were immobilized in CELLlview slides (Greiner bio-one) by treating wells beforehand with 30 μl 0.1% poly-L-lysine, then removing poly-L-lysine by aspiration and hydrating wells with a drop of water. Thereafter, 10 μl of cells was added to the well and let to sit for 10 mins. Cells in suspension were aspirated and the remain coated cells dried for 1 min. Finally, a drop of SlowFade Gold Antifade Mountant (Thermo Fisher) was added to wells containing immobilised cells.

## Protein purification and production of polyclonal antibodies

In order to immunize rabbits for production of ChvI polyclonal antibodies His6-ChvI was purified as follows. A BL21(DE3) strain harboring plasmid pET-28a-*chvI* was grown in LB medium supplemented with kanamycin until an $OD_{600}$ of 0.7 was reached. IPTG (isopropyl—D-thiogalactopyranoside) (Thermo Fisher Scientific) was added at a final concentration of 1 mM, and the culture was incubated overnight at 18°C. Then, cells were harvested by centrifugation for 30 min at 4,000 rpm and 4°C. The pellet was resuspended in 20 ml BD buffer (20 mM Tris-HCl [pH 8.0], 500 mM NaCl, 10% glycerol, 10 mM MgCl2, 12.5 mM imidazole) supplemented with complete EDTA-free protease cocktail inhibitor (Roche), 400 mg lysozyme (Sigma), and 10 mg DNase I (Roche) and incubated for 30 min on ice. Cells were then lysed by sonication and the lysate cleared by centrifugation (12,000 rpm for 30 min at 4°C) was loaded on a Ni-nitrilotriacetic acid (Ni-NTA) column and incubated for 1 h at 4°C with end-over-end shaking. The column was then washed with 5 ml BD buffer, 3 ml Wash 1 buffer (BD buffer with 25 mM imidazole), 3 ml Wash 2 buffer (BD buffer with 50 mM imidazole), and 3 ml Wash 3 buffer (BD buffer with 75 mM imidazole). Proteins bound to the column were eluted with 3 ml elution buffer (BD buffer with 100 mM imidazole) and aliquoted in 300 μl fractions. All the fractions containing the protein of interest (checked by Coomassie blue staining) were pooled and dialyzed in dialysis buffer (20 mM Tris [pH 8.0], 0.5 M NaCl, 10% glycerol). Purified ChvI was used to immunize rabbits (CER Groupe, Marloie, Belgium).

## Immunoblot analysis

Proteins crude extracts were prepared by harvesting cells from exponential growth phase ($OD_{660}$ ~ 0.3). The pellets were then resuspended in SDS-PAGE loading buffer by normalizing to the $OD_{660}$ before lysing cells by incubating them for 10 min at 95°C. 20 μl of cell lysates was loaded in wells of a 12% SDS-polyacrylamide gel for protein separation by electrophoresis. Thereafter, proteins were transferred onto a nitrocellulose membrane then blocked overnight in 5% (wt/vol) non-fat dry milk in phosphate buffer saline (PBS) with 0.05% Tween 20. Membrane was immunoblotted for $\geq$ 3 h with primary monoclonal anti-GFP (1:5,000) antibodies (JL8, Clontech-Takara), then followed by immunoblotting for $\leq$ 1 h with secondary antibodies: 1:5,000) anti-mouse linked to peroxidase (Dako Agilent), and vizualized thanks to Clarity™ Western ECL substrate chemiluminescence reagent (BioRad) and Amersham Imager 600 (GE Healthcare).

## Chromatin immunoprecipitation followed by deep sequencing (ChIP-Seq) assay

A ChIP-Seq protocol was followed as described in [57]. Briefly, 80 ml of mid-log-phase cells ($OD_{660}$ of 0.6) were cross-linked in 1% formaldehyde and 10 mM sodium phosphate (pH 7.6) at room temperature (RT) for 10 min and then for 30 min on ice. Cross-linking was stopped by addition of 125 mM glycine and incubated for 5 min on ice. Cells were washed twice in phosphate buffer solution (PBS; 137 mM NaCl, 2.7 mM KCl, 10 mM $Na_2HPO_4$, 1.8 mM $KH_2PO_4$, pH 7.4) resuspended in 450 μl TES buffer (10 mM Tris-HCl [pH 7.5], 1 mM EDTA, and 100 mM NaCl), and lysed with 2 μl of Ready-lyse lysozyme solution for 5 min at RT. Protease inhibitors (Roche) were added, and the mixture was incubated for 10 min. Then, 550 μl of ChIP buffer (1.1% Triton X-100, 1.2 mM EDTA, 16.7 mM Tris-HCl [pH 8.1], and 167 mM NaCl, plus protease inhibitors) were added to the lysate and incubated at 37°C for 10 min before sonication (2 x 8 bursts of 30 sec on ice using a Diagenode Bioruptor) to shear DNA fragments to a length of 300 to 500 bp. Lysate was cleared by centrifugation for 10 min at

12,500 rpm at 4°C, and protein content was assessed by measuring the $OD_{280}$. Then, 7.5 mg of proteins was diluted in ChIP buffer supplemented with 0.01% SDS and precleared for 1 h at 4°C with 50 µl of SureBeads Protein A Magnetic Beads (BioRad) and 100 µg bovine serum albumin (BSA). One microliter of polyclonal anti-ChvI antibodies was added to the supernatant before overnight incubation at 4°C under gentle agitation. Next, 80 µl of BSA presaturated protein A-agarose beads were added to the solution and incubated for 2 h at 4°C with rotation, washed once with low-salt buffer (0.1% SDS, 1% Triton X-100, 2 mM EDTA, 20 mM Tris-HCl [pH 8.1], 150 mM NaCl), once with high-salt buffer (0.1% SDS, 1% Triton X-100, 2 mM EDTA, 20 mM Tris-HCl [pH 8.1], 500 mM NaCl), once with LiCl buffer (0.25 M LiCl, 1% NP-40, 1% deoxycholate, 1 mM EDTA, 10 mM Tris-HCl [pH 8.1]), and once with TE buffer (10 mM Tris-HCl [pH 8.1] 1 mM EDTA) at 4°C, followed by a second wash with TE buffer at RT. The DNA-protein complexes were eluted twice in 250 µl freshly prepared elution buffer (0.1 M NaHCO3, 1% SDS). NaCl was added at a concentration of 300 mM to the combined eluates (500 µl) before overnight incubation at 65°C to reverse the cross-link. The samples were treated with 20 µg of proteinase K in 40 mM EDTA and 40 mM Tris-HCl (pH 6.5) for 2 h at 45°C. DNA was extracted using a Nucleospin PCR cleanup kit (Macherey-Nagel) and resuspended in 50 µl elution buffer (5 mM Tris-HCl [pH 8.5]). DNA sequencing was performed using an Illumina NextSeq 550 (paired-end 2x75) instrument (BIO.be). NGS data were analysed as described in [57].

## RNA-seq

WT and Δ*chvI* cells were grown overnight to $OD_{660}$ ~ 0.3 and then exposed to 6% sucrose for 4 h. Thereafter, total RNA was extracted with RNeasy Protect Bacteria Kit from Qiagen and following manufacturer's instructions. The quantity and quality (A260/A280 ratio) of RNA was determined with a Thermo Scientific Nanodrop One Microvolume UV-Vis Spectrophotometer. RNASeq TTRNA libraries were prepared according to the manufacturer's instructions and sequenced with Illumina NovaSeq 6000 (paired-end 2x100) instrument (BIO.be). NGS data were analysed using Galaxy (https://usegalaxy.org) [58]. Briefly, FastQC was used to evaluate the quality of the reads; HISAT2 was used to map the reads onto the NA1000 reference genome (NC_011916.1) and generate bam files; featureCounts was used to generate counts tables using bam files and DESeq2 was used to determine differentially expressed genes. The Volcano plot was generated using GraphPad Prism 9 software.

## MEME analysis

Sequences corresponding to +/- 100bp from TSS of genes identified in the ChIP-seq and reported as either positive or negative regulated from the RNA-seq and β-galactosidase activity from promoter fusions were analysed for a ChvI conserved DNA binding motif using MEME [59] (**S3 Table**). As searching parameters, the classical motif discovery mode, zoops motif site distribution, 10 minimum motif width and 15 maximum motif width and a p-value<0.001 as cut-off were used.

## *In vivo* ³²P labelling

Cells were grown overnight in PYE, then washed twice and grown in M5GG medium overnight in M5G with 0.05 mM phosphate to $OD_{660}$ ~ 0.3. Immediately before reaching the desired OD for cultures, 40 µl of SureBeads Protein A Magnetic Beads (BioRad) were washed 4 times in 1 ml PBS + 0.1% Tween (PBS-T) and finally resuspended in 65 µl. Thereafter, 3 µl of anti-ChvI antibody were added to the washed protein A-agarose beads and incubated at RT in a shaker at 1300 rpm for 30 mins. Protein A beads with anti-ChvI antibodies were washed 3

times with PBS-T and 1 time with cold (4˚C) Low Salt Buffer (LSB; 50 mM Tris pH 7.0, 100 mM NaCl, 50 mM EDTA, 2% Triton X100; sterilised with 2 μm Acrodisc syringe filters), immediately resuspended in 50 μl LSB and kept on ice. Then, one ml of cell culture was labelled using 30 μCi γ-[$^{32}$P]ATP (PerkinElmer). Samples were immediately incubated with either 6% sucrose, 10 μg/ml vancomycin or 100 μg/ml mecillinam for 7 minutes at 30˚C. Then, cells were pelleted for 2 min at 15,000 rpm and the supernatant removed completely without disturbing the pellets. Cell pellets were resuspended in 50 μl Lysis Buffer (50 mM Tris pH 7.0, 150 mM NaCl, 80 mM EDTA, 2% Triton X100; sterilised with 2 μm Acrodisc syringe filters) by pipetting. Lysed samples were incubated 3 min on ice, 450 μl of cold LSB were added and proteins were collected using centrifugation for 15 min at 13,000 and 4˚C. Then, ChvI immunoprecipitation (IP) was performed by adding the Protein A beads with anti-ChvI antibodies previously prepared to the cell-free protein samples, and incubating at 4˚C with rotation for 90 minutes. IP samples were washed 1 time in 1 ml LSB and 3 times in 1 ml High Salt Buffer (50 mM Tris pH 7.0, 500 mM NaCl, 50 mM EDTA, 0.1% Triton X100; sterilised with 2 μm Acrodisc syringe filters). Samples were eluted using 25 μl of 2.5X SDS-loading Buffer and incubation at 37˚C for 5 min. Samples with SDS loading Buffer were separated from magnetic beads, and 20 μl loaded on a SDS-Page gel and run at 200 V-50 mA-100 W. The gel was dried for 1h at 70˚C under vacuum in a Model 583 Gel Dryer (BioRad). Finally, the dried gel was exposed on phosphoscreen for 5 days and revealed using a Cyclone Plus Phosphor Imager (PerkinElmer).

### Quantification of ChvI phosphorylation

ChvI phosphorylation levels were obtained following the method for quantification of western blots in FIJI. Briefly, intensities of phosphorylated protein bands in phosphoscreens were analysed from images taken in the Cyclone Plus Phosphor Imager (PerkinElmer). TIFF images were converted to jpeg format and to grayscale picture mode. The "grey mean value" was set as criteria for measurement and a fixed size rectangle large enough to contain the largest band in the image was used to frame the protein samples across a row. Bands were centered in the fixed size rectangle and the grey mean value taken to each protein band using the measurement option. Background corrected absolute values were obtained by measuring a portion of the gel that had no protein samples loaded and the band intensities subtracted from them.

### Statistical analyses

All the statistical analyses were performed using GraphPad Prism 9 software. A *P* value of <0.05 was considered as significant.

### Supporting information

**S1 Fig. Growth and morphology of *chvI* mutant strains in synthetic minimal media.** (A) Growth of WT, *chvID52E* and *chvID52A* mutant strains in M2G. The data represent the average values of biological replicates (n = 3, error bars show standard deviation). (B) Morphology of WT, Δ*chvI*, *chvID52E* and *chvID52A* cells after 48 hrs of incubation in M2G. Scale bars in microscopy images correspond to 1 μm.
(EPS)

**S2 Fig. The promoter activity of genes studied from the ChvI regulon are upregulated upon different osmotic regimes.** (A) Activity of the *dipM*, *ftsN*, *nepR*, *phyR* and *chvI* promoters (Miller Units) in WT (black bars) and Δ*chvI* (white bars) cells grown overnight in PYE, then washed and exposed for 4 h in M2G. (B-D) Activity of the *dipM* (A), *ftsN* (B) and *phyR*

(C) promoters in WT cells treated with different osmotic conditions. Seed cells were grown in PYE and exposed to the different conditions for four hours. Data in M2G was obtained from cells pelleted from PYE cultures, washed twice and exposed to M2G. The data represent the average values of biological replicates (n = 3, error bars show standard deviation). $**$ = $p < 0.01$, NS = $p > 0.05$, Single factor ANOVA analysis of β-galactosidase activity.
(EPS)

**S3 Fig. *chvI* and *sigT* are interconnected.** (A) Activity of the *chvI* promoter P*chvI* (in Miller Units) in WT (black bars), Δ*chvI* (light grey bars) Δ*sigT* (dark grey bars) cells grown in PYE and exposed to different osmotic conditions. Data in M2G was obtained from cells pelleted from PYE cultures, washed twice and exposed to M2G. The data represent the average values of biological replicates (n = 3, error bars show standard deviation). $**$ = $p < 0.01$, NS = $p > 0.05$, Single factor ANOVA analysis of β-galactosidase activity. (B) Viability of single Δ*chvI* and Δ*sigT* mutants in PYE supplemented with different osmolytes at different concentrations. Images are representative of three biological replicates.
(EPS)

**S4 Fig. Δ*chvI* is not sensitive to every cell envelope stress.** (A) Growth of WT (black) and Δ*chvI* (orange) cells in PYE with (dashed lines) or without (solid lines) acidic stress pH 5.5, polymixin B (Pol B) and cephalexin (Ceph). The data represent the average values of biological replicates (n = 3, error bars show standard deviation). (B) Viability of Δ*chvI* cells on plates supplemented with 0.001% or 0.0025% sodium dodecyl sulfate (SDS), Pol B, Ceph or A22. Images are representative of three biological replicates. (C) Viability of single (Δ*chvI* and Δ*chvT*) and double (Δ*chvI* Δ*chvT*) mutants on PYE agar with vancomycin (Van), mecillinam (Mec), cefsulodin (Cef) and moenomycin (Moe). Images represent three biological replicates.
(EPS)

**S5 Fig. ChvG relocates from a patchy-spotty pattern to mid-cell upon osmotic upshift.** (A) Growth of Δ*chvI* and Δ*chvG* mutants complemented with *chvI* and *chvG* N- and C-terminal eGFP fusions in M2G supplemented with xylose. (B) Immunodetection of eGFP fusions with GFP antibodies. The expected molecular weights for eGFP, eGFP-ChvI, ChvI-eGFP, eGFP-ChvG and ChvG-eGFP are 26.94, 54.33, 55.75, 85.88, 86.06 KDa, respectively. (C) Localisation of eGFP-ChvG in a Δ*chvG* background grown overnight in PYE and M2G and imaged in PYE and M2G agarose pads, respectively. (D) Confocal microscopy images of Δ*chvG* cells expressing eGFP-ChvG cells grown in PYE. (E-F) Growth upon endogenous (E) and ectopic expression (F) of *chvGH309N* in M2G. (G) Impact of deletions in sensor (Δ*chvG*274-534) and catalytic (Δ*chvG*1-274) domains of ChvG. (H) Growth of Δ*chvG* cells expressing either full length or truncated portions of ChvG fused to reporters. The data represent the average value of biological replicates (n = 3, error bars show standard deviation), except for panel A. Expression of eGFP fusions from P*xylX* was induced with 0.1% xylose.
(EPS)

**S6 Fig. Replicates of 1A, 2, 4C, 5E, 5F, S3B and S3C Figs.**
(PDF)

**S7 Fig. Numerical data and statistics.**
(XLSX)

**S1 Table. ChvI direct targets determined by ChIP-seq.** Promoter regions bound by ChvI in WT cells grown in synthetic minimal media M2G.
(PDF)

**S2 Table. ChvI regulon determined by RNA-seq.** (A) genes with expression down-regulated or (B) up-regulated in the Δ*chvI* mutant upon osmotic stress with 6% sucrose.
(PDF)

**S3 Table. Sequences for ChvI MEME analysis.**
(PDF)

**S4 Table. Bacterial strains used in this study.**
(PDF)

**S5 Table. Plasmids used in this study.**
(PDF)

**S6 Table. Oligonucleotides used in this study.**
(PDF)

**S1 Text. Supplementary methods.** Construction of plasmids.
(PDF)

## Acknowledgments

We are grateful to Sean Crosson and Benjamin Stein for willingness to discuss and share data on their ChvGI-related projects. Also, we would like to thank members of the URBM of the University of Namur for their comments and suggestions on this project, especially Dr. Angéline Reboul for her comments and advices on the microscopy analysis, Jérôme Coppine for plasmids constructs and his help with phosphorylation assays, and members of the Hallez lab for their comments on the manuscript.

## Author Contributions

**Conceptualization:** Alex Quintero-Yanes, Régis Hallez.

**Data curation:** Alex Quintero-Yanes.

**Formal analysis:** Alex Quintero-Yanes, Aurélie Mayard.

**Funding acquisition:** Régis Hallez.

**Investigation:** Alex Quintero-Yanes, Aurélie Mayard.

**Methodology:** Alex Quintero-Yanes.

**Project administration:** Régis Hallez.

**Resources:** Régis Hallez.

**Supervision:** Régis Hallez.

**Validation:** Alex Quintero-Yanes.

**Visualization:** Alex Quintero-Yanes.

**Writing – original draft:** Alex Quintero-Yanes, Régis Hallez.

**Writing – review & editing:** Régis Hallez.

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
