## [Decision Letter · Decision Letter 0]

5 Jul 2022

Dear Dr Hallez,

Thank you very much for submitting your Research Article entitled 'The two-component system ChvGI maintains cell envelope homeostasis in Caulobacter crescentus' to PLOS Genetics.

The manuscript was fully evaluated at the editorial level and by independent peer reviewers. The reviewers appreciated the attention to an important problem, but raised some substantial concerns about the current manuscript. Based on the reviews, we will not be able to accept this version of the manuscript, but we would be happy to review a much-revised version.

Should you decide to submit a revised manuscript it is important that you address experimentally #1 raised by reviewer 1. Also, as pointed out by reviewer 2, please include uncropped blots and compare and contrast ChvGI transcriptome data from Caulobacter to those of Sinorhizobium and Agrobacterium. Moreover, your revisions should address the other specific points made by each reviewer. We will also require a detailed list of your responses to the review comments and a description of the changes you have made in the manuscript.

If you decide to revise the manuscript for further consideration at PLOS Genetics, please aim to resubmit within the next 60 days, unless it will take extra time to address the concerns of the reviewers, in which case we would appreciate an expected resubmission date by email to plosgenetics@plos.org.

[LINK]

We are sorry that we cannot be more positive about your manuscript at this stage. Please do not hesitate to contact us if you have any concerns or questions.

Yours sincerely,

Lotte Søgaard-Andersen, Ph.D., M.D.

Section Editor: Prokaryotic Genetics

PLOS Genetics

Lotte Søgaard-Andersen

Section Editor: Prokaryotic Genetics

PLOS Genetics

Reviewer's Responses to Questions

**Comments to the Authors:**

Reviewer #1: All comments on the manuscript are in the attachment.

Reviewer #2: Summary:

This manuscript from Quintero-Yanes et al. reports analysis of the ChvG-ChvI two component system in Caulobacter crescentus, providing a number of new observations about the conditions in which it is required for cell viability, its regulon, phosphorylation and cellular localization of the regulators.

Major Comments:

This study adds significantly to the understanding of the ChvGI regulators in C. crescentus. The experiments are wide-ranging and generate a wealth of data, that expands on prior studies in C. crescentus.

The introduction does a nice job of succinctly providing context for the study, however I find that the transcriptome analysis does not take advantage of prior studies in other APB to compare the Cc regulon to those of other systems. The ChvG-ChvI (ExoR) transcriptomic data is available for S. meliloti and A. tumefaciens and could be productively compared. These systems are activated by acidic pH through proteolytic cleavage of the ExoR protein and it is a shame not to compare and contrast the Cc system in which no such regulator has been identified. I also find the comparison between the ChIP-Seq and RNASeq within this study to be quite minimal. How strong is the agreement? How much of the ChvGI regulon is directly vs indirectly impacted?

Overall, the fold-changes of most target genes and the impacts of the regulators seem modest, often less than two-fold for the beta-gal fusions. Is there any explanation why this might be? Caulobacter has a biphasic life cycle so perhaps expression is biased towards one cell type and the total amount of expression normalized to total biomass is an underrepresentation? Even so, this would likely only have a two-fold effect at most. How do these levels compare to other regulatory systems in Cc?

Minor Comments:

1. L53 – “its” rather than his

2. L70-1. The proteolysis of ExoR was also reported for S. meliloti – Lu et al. 2012, JB 194:4029. In fact, the proteolysis was reported first in S. meliloti and cited in the Wu et al. paper.

3. L106. “We provide a ChvI regulon…” should be “We provide information on the ChvI regulon….” Or we elucidate the……” ChvI regolun

4. L146. This is PYE + 50 mm KCl in the figure.

5. L207-8. Why were the conditions in PYE modified from those used in the expression and resistance assays to PYE + 6% sucrose. Some explanation of this shift is warranted.

6. L213-4. Is there any explanation for the incongruity between the RNA-Seq and the fusions for ftsN and phyR?

7. L229. AND ALSO THOSE ENCODING the elongasome components…– otherwise, this can be interpreted as other genes/proteins affected by StaR.

8. L248-9. Given that it was the chvG mutant that had to be tested for chvI phosphorylation it would be more appropriate to document the growth of the chvG mutant in the M5GG in 4A. Was the chvG growth phenotype identical to that of the chvI mutant?

9. L250-53. It needs to be made clear here that these results are immunoprecipitation reactions with anti-ChvI Abs. As described in the results this was quite unclear and I found myself questioning how these in vivo labeling reactions were so clean. Also should be mentioned in the legend for Fig. 4. Also, there is no internal control and with limited details on how these were normalized. For an example where this could be problematic, the ChvI-P in the wild type with no sucrose in panels B and C is not comparable in intensity.

10. L266. The polymyxin B treatment seems to be right on the edge of significance. There does seem to be a consistent difference in growth.

11. L283-5. The difference in timing would be more convincing presented as a time course.

12. L319. In C. crescentus.

13. L339. Was there any logic to why the sensor domain was truncated at pos. 114? Seems an odd truncation site.

14. L401. OF the SigT-dependent regulon

15. L456. How do the authors propose that ChvG senses the pressure? Presumably they mean osmotic pressure. Are there comparable systems in other bacteria. Would this be similar to EnvZ-dependent sensing in E. coli?

16. L471. Should provide a reference for the statement about MreB orthologs.

17. L493-4. This reads confusingly – Perhaps – “..under conditions in which the cell envelope is unperturbed compared to conditions which compromise the envelope.”?

18. L982. How are these extracts normalized?

Reviewer #3: This manuscript focuses on the two-component ChvIG systems conserved in a number of alphaproteobacteria. Mutants in this TCS are known to show severe growth defects in synthetic minimal media while they grow similarly to the WT in complex medium. The main findings of this new study are (i) that the growth defect is suppressed by high concentrations of osmolytes in the medium, (ii) that phosphorylation of the response regulator ChvI is induced by osmotic upshift and antibiotics targeting peptidoglycan synthesis, (iii) that the receptor kinase ChvG shows a patchy cellular localization pattern and forms foci in the mid cell area upon transition from complex to synthetic medium resembling localization patterns and dynamics of peptidoglycan-related proteins, (v) confinement of ChvI domains required for this localization pattern, and (v) a further confinement of the ChvI regulon based on ChIP-seq data which in correlation with RNA-seq transcriptome data suggest direct gene targets of ChvI regulation. Altogether the manuscript provides a conglomeration of new information on factors activating this important TCS and further evidence for its role in cell envelope homeostasis under cell envelope stress conditions. This study is poor in providing mechanistic insights into signal perception by the ChvGI system but provides some starting points (particularly the observed cellular localization patterns) for future studies towards a deeper understanding of the underlying molecular mechanisms.

Although some improvement is required the manuscript is well written and conclusions are fully supported by the presented data.

Major suggestion/concerns

Were the peak regions identified by ChIP-seq analyzed for a consensus sequence that may represent the ChvI binding site and if putative binding site sequences can be identified, is there a correlation between positive or negative regulation by ChvI with the position of the binding site relative to promoter motifs and transcription start sites?

Fig. 4B and C, 5F: Was a loading control used to guarantee that the same amount of protein was loaded? This is considered important for the quantitative comparisons.

Minor comments

l. 67-68: ”pathogenic regulator” should be replaced by ‘pathogenicity regulator’ or regulator of pathogenicity”

l. 136 “… whereas expressing back chvI” Please rephrase.

ll. 137-138: Please add references for the phosphor-mimetic and phosphor-ablative chvI variant, as a proof for the properties of these protein variants is important, but provided in a previous study.

ll. 166-168: “This suggests either that ChvID52E does not perfectly mimic phospohorylated …” Yet another possibility is that the level of active ChvI is too high, as every protein of the phosphor-mimetic variant is probably active and cannot be inactivated by dephosphorylation.

ll. 191-192: “… the chvI promoter (PchvI) was significantly higher …” should read ““… the chvI promoter (PchvI) activity was significantly higher …”

ll. 196-198: “To better understand the regulatory mechanism between ChvI and SigT we tested viability of mutants for the genes in osmotic conditions used in this study, …” Please rephrase for clarity. It is not clear what is meant by “mutants for genes in osmotic conditions”.

l. 198, l. 200: “… that is PYE supplemented …” should read “… that is PYE supplemented with …”

ll. 202-203: “… the viability of �sigT declined.” should read “… the viability of �sigT mutant declined.”

ll. 216-217: “… dipM and chvI promoters activity …” should read “… dipM and chvI promoter activity …”

ll. 245-246: “… which is much less that in M2G …” should read “… which is much less than in M2G …”

Fig. 1A, Fig. 2, Fig. 5E: spotting assays: how many replicates are represented by the displayed images? Results from further replicates should be provided in a data repository.

Fig. 1B-D, Fig. 4A, Fig. 5A-D: y-axis – replace commas used as decimal separator

Fig. 2: The two further replicates not shown should be included in the supplement or in a data repository.

References in reference list require further formatting.

**Have all data underlying the figures and results presented in the manuscript been provided?**

Reviewer #1: Yes

Reviewer #2: **No: **Data for most graphs was not included. Also, the gel for 32P labeling experiment was highly cropped and the original was not provided as back-up. Large scale data was deposited in appropriate repositories.

Reviewer #3: Yes

PLOS authors have the option to publish the peer review history of their article (what does this mean?). If published, this will include your full peer review and any attached files.

Reviewer #1: No

Reviewer #2: No

Reviewer #3: No

---

## [Decision Letter · Decision Letter 1]

30 Sep 2022

Dear Dr Hallez,

Thank you very much for submitting your Research Article entitled 'The two-component system ChvGI maintains cell envelope homeostasis in Caulobacter crescentus' to PLOS Genetics.

The manuscript was fully evaluated at the editorial level and by independent peer reviewers. The reviewers appreciated the attention to an important topic but identified  a few minor concerns that can be addressed by careful editing.

We therefore ask you to modify the manuscript according to the review recommendations. Your revisions should address the specific points made by each reviewer.

Yours sincerely,

Lotte Søgaard-Andersen, Ph.D., M.D.

Section Editor

PLOS Genetics

Reviewer's Responses to Questions

**Comments to the Authors:**

Reviewer #1: The authors have made thorough revisions to their manuscript which address all my prior comments and experimentally strengthen the conclusions that they draw from the data. The manuscript is noticeably improved as a result. I have only a few minor comments on the revised version, below.

Minor comments

Line 128, for clarity: “viability of chvIG mutants on M2G plates lacking Na+ and K+ salts

Line 212, remove “for”

Lines 250-252, while it doesn’t invalidate the authors conclusions, I don’t really see much difference in viability between WT and the sigT mutant strain in any condition tested, including the most stressful ones (Fig S3B). In some of the replicates in the supplemental data it is clearer, but still weak. Suggest “the effect of these osmotic stressors on the sigT mutant was minimal” or similar phrasing. I don’t think the authors need to make any more experiments, as the effect of sigT loss during osmotic stress has been characterized already in other work.

Line 346, remove “to”

Line 451, “higher salt concentrations” (remove extra “s”)

Line 479, based on the data presented, the authors haven’t directly observed that the antibiotics cross the outer membrane at different rates, it is still a hypothesis. Suggest “deduce” instead of “observe” here.

Reviewer #2: Quintero-Yanes et al. have extensively revised their manuscript from the prior submission, and have largely addressed my initial comments, and most of those from the other reviewers. For the most part the manuscript has been substantially improved.

The section on the identified ChvI binding site remains a little fuzzy, and could be cleared up. I think the authors should add the binding site map they present in their responses to the supplemental figures. Also, they should include the previously published sequence for the ChvI binding site from S. meliloti, so that their sequence from C. crescentus can be compared to what was proposed before. I am more inclined to believe the binding site identified here based on the ChIP-Seq analysi, not to mention that fact that they are comparing this to a different genus of bacteria in a different Order. I do agree that experimental validation is also needed from this analysis, but the current manuscript already represents a large study and I am not suggesting that this validation should be included here.

There are few other minor issues in this section as well.

L414 - consensus ChvI bonding sequence should be "ChvI binding sequence"

L416 - The phrase "....for the ChvI showed a reasonable conserved DNA binding motif..." is confusing. It sounds as if they have defined a DNA binding motif on ChvI. Rephrasing to read "...revealed reasonably conserved DNA sequences that could be recognized by the ChvI protein...."

**Have all data underlying the figures and results presented in the manuscript been provided?**

Reviewer #1: Yes

Reviewer #2: **No: **The numerical data underlying information presented in graphical form has not been provided (or I could not find it), although the authors have uploaded large scale data to appropriate databases and now include full mages of their gels.

PLOS authors have the option to publish the peer review history of their article (what does this mean?). If published, this will include your full peer review and any attached files.

Reviewer #1: No

Reviewer #2: No

---

## [Editor Report · Decision Letter 2]

9 Oct 2022

Dear Dr Hallez,

We are pleased to inform you that your manuscript entitled "The two-component system ChvGI maintains cell envelope homeostasis in Caulobacter crescentus" has been editorially accepted for publication in PLOS Genetics. Congratulations!

Yours sincerely,

Lotte Søgaard-Andersen, Ph.D., M.D.

Section Editor

PLOS Genetics

Comments from the reviewers (if applicable):

**Data Deposition**

http://datadryad.org/submit?journalID=pgenetics&manu=PGENETICS-D-22-00606R2

**Press Queries**

---

## [Editor Report · Acceptance letter]

19 Oct 2022

PGENETICS-D-22-00606R2 

The two-component system ChvGI maintains cell envelope homeostasis in Caulobacter crescentus 

Dear Dr Hallez, 

We are pleased to inform you that your manuscript entitled "The two-component system ChvGI maintains cell envelope homeostasis in Caulobacter crescentus" has been formally accepted for publication in PLOS Genetics! Your manuscript is now with our production department and you will be notified of the publication date in due course.

With kind regards,

Zsofi Zombor

PLOS Genetics

On behalf of:
